# Discarded sequencing reads uncover natural variation in pest resistance in *Thlaspi arvense*

**Dario Galanti[1,2]\*, Jun Hee Jung[1], Caroline Müller[3], Oliver Bossdorf[1]\***

[1]Plant Evolutionary Ecology, Institute of Evolution and Ecology, University of Tübingen, Tübingen, Germany; [2]Royal Botanic Gardens, Kew, Richmond upon Thames, United Kingdom; [3]Chemical Ecology, Bielefeld University, Bielefeld, Germany

## eLife Assessment

This **important** study presents a significant methodological advance by leveraging previously discarded, unmapped DNA sequence reads to estimate pest infestation loads across plant accessions, and map variation in these apparent pest loads to defense genes. The bioinformatics approach is **compelling**, and the results should bear broad implications for phenotype-genotype prediction, especially regarding the use of unmapped reads for GWAS.

**\*For correspondence:**
dario.galanti@uni-tuebingen.de (DG);
oliver.bossdorf@uni-tuebingen.de (OB)

**Competing interest:** The authors declare that no competing interests exist.

**Abstract** Understanding the genomic basis of natural variation in plant pest resistance is an important goal in plant science, but it usually requires large and labor-intensive phenotyping experiments. Here, we explored the possibility that non-target reads from plant DNA sequencing can serve as phenotyping proxies for addressing such questions. We used data from a whole-genome and -epigenome sequencing study of 207 natural lines of field pennycress (*Thlaspi arvense*) that were grown in a common environment and spontaneously colonized by aphids, mildew, and other microbes. We found that the numbers of non-target reads assigned to the pest species differed between populations, had significant SNP-based heritability, and were associated with climate of origin and baseline glucosinolate contents. Specifically, pennycress lines from cold and thermally fluctuating habitats, presumably less favorable to aphids, showed higher aphid DNA load, i.e., decreased aphid resistance. Genome-wide association analyses identified genetic variants at known defense genes but also novel genomic regions associated with variation in aphid and mildew DNA load. Moreover, we found several differentially methylated regions associated with pathogen loads, in particular differential methylation at transposons and hypomethylation in the promoter of a gene involved in stomatal closure, likely induced by pathogens. Our study provides first insights into the defense mechanisms of *Thlaspi arvense*, a rising crop and model species, and demonstrates that non-target whole-genome sequencing reads, usually discarded, can be leveraged to estimate intensities of plant biotic interactions. With rapidly increasing numbers of large sequencing datasets worldwide, this approach should have broad application in fundamental and applied research.

## Introduction

Plant pests, such as pathogens and herbivores, can cause major yield losses in crops and often require the massive use of pesticides to control their damage. Natural plant populations, on the other hand, are constantly exposed to such biotic stressors and their higher genetic diversity often allows these populations to become locally adapted. Since many pest species are sensitive to climatic conditions, their pressure on plant communities is spatially heterogeneous, maintaining geographically structured

**eLife digest** The genetic code of organisms is made of DNA, a molecule consisting of long sequences of four different base pairs. To gain insights into the organisms' genetic information, it is necessary to establish which base pairs are in its DNA and in what order. This is known as 'sequencing', and it allows scientists to 'read out' the genetic information of an organism.

Technically, sequencing often involves shearing the organisms' DNA into smaller pieces, so that the enzymes that do the sequencing can fully 'read' each molecule of DNA. However, when DNA is isolated from an organism, for example a plant, not only the DNA from the plant will be obtained. A small portion of DNA from other organisms, including viruses, bacteria, fungi and even insects that visited the plant will also be isolated and sequenced. These 'non-target' DNA fragments are usually discarded because they do not match the reference genome of the sequenced plant.

However, the genetic information of these other organisms can provide additional insights into the plant. This is particularly true when scientists sequence a large collection of individual plants from the same species. In this case, the DNA of other organisms isolated along with each plant's own DNA can tell researchers about differences between the plants, such as whether they are able to resist a particular disease or establish symbiosis with a specific fungus.

Galanti et al. wanted to find out more about the genetic background and characteristics of a European plant called the field pennycress, *Thlaspi arvense*. To do this, they used the fact that plants from different regions would acquire different pests depending on their genetic background, and the fact that the DNA from different creatures living with the plant would be gathered when the plant DNA was collected.

First, Galanti et al. collected pennycress seeds from across Europe and grew them in the same environment, and then they let these plants be colonized by pests. Next, the researchers tested whether the DNA of pests living on the plants reflected differences in resistance to these pests, and whether that could explain why some plants were more or less resistant based on their geographic origin and genetic background.

Galanti et al. found that, in general, plants collected in warmer and thermally stable climates, where pests usually thrive, had fewer pests in the controlled environment, suggesting that these plants had developed resistance to the pests. With this information, the researchers were also able to unravel the genetic bases of resistance, finding genetic variants in the plants with pests that were close to defense genes.

These results highlight the potential of acquiring important insights from non-target DNA fragments, especially to study plant-pathogen interactions. This could be useful in plant breeding programmes.

genetic variation in plant defenses (*Züst et al., 2012*; *Kerwin et al., 2015*). For these reasons, natural plant populations are highly suitable to study defense mechanisms and evolution of defenses, and also a very useful source of beneficial and resistance alleles for specific pathogens and environmental conditions. This genetic variation in defense-related genes can for example be screened through genome-wide association (GWA) (*Chan et al., 2010*; *Corwin et al., 2016*; *Thoen et al., 2017*; *Hanson et al., 2018*) or approaches based on known candidate genes (*Kerwin et al., 2015*).

Many pest species are also highly sensitive to temporal variation in weather conditions. This temporal heterogeneity in pathogen pressure can induce plastic responses in plants, involving gene expression and epigenetic changes (*Jaouannet et al., 2015*; *Geng et al., 2019*; *Annacondia et al., 2021*), which may also be studied through stress experiments (*Jaouannet et al., 2015*; *Geng et al., 2019*; *Annacondia et al., 2021*). Some plastic epigenetic responses can have a transient stability and be transmitted to the next generations through inheritance of epigenetic marks (*Kinoshita and Seki, 2014*; *Espinas et al., 2016*; *Lämke and Bäurle, 2017*; *He and Li, 2018*). In particular, DNA methylation has been shown to respond to biotic and abiotic stresses through gene expression regulation and transposable elements (TEs) reactivation, and can be inherited across generations (*Annacondia et al., 2021*; *Lämke and Bäurle, 2017*; *Roquis et al., 2021*). In plants, DNA methylation can occur in the three sequence contexts CG, CHG, and CHH (H being A, T, or C), which differ in their molecular machineries depositing, maintaining, and removing methylation and consequently also in their

transgenerational stability (*Law and Jacobsen, 2010*; *Zhang et al., 2018*). While CG methylation is usually more stable across generations, CHH methylation is less stable and more responsive to stress and the sensitivity of CHG methylation lies somewhere in between (*Law and Jacobsen, 2010*; *Zhang et al., 2018*; *Liu and He, 2020*).

Whether inherited or induced, some strategies of plants for defense against pathogens and herbivores include: (i) physical barriers such as reinforced cell walls, leaf protective layers, or closing stomata, (ii) production of specialized (secondary) metabolites that reduce palatability or are toxic to pests, (iii) oxidative bursts, (iv) the activation of signaling cascades to induce systemic responses, and (v) RNA interference mechanisms to silence pathogen genes (*Wojtaszek, 1997*; *War et al., 2012*; *Kant et al., 2015*; *Melotto et al., 2017*; *Muhammad et al., 2019*). In Brassicaceae, a particularly important and diverse class of defense metabolites are glucosinolates, which often show local adaptation driven by variation in pests and can also be induced by herbivore and pathogen attacks (*Züst et al., 2012*; *Kerwin et al., 2015*; *Kutyniok and Müller, 2012*).

Studying natural variation in plant resistance, along with associated genetic and epigenetic variation, can identify genes involved in defense and their regulators, including vital genes whose function cannot be determined through knockout experiments. Such knowledge, and especially the discovery of natural resistance alleles, are crucial sources for the breeding of more pest-resistant crop varieties. Nevertheless, because of the diversity of resistance mechanisms and their often multigenic nature, plant defense mechanisms remain difficult to study. In particular, antixenosis (the prevention of pathogen settlement) and antibiosis (the repression of pathogen growth and reproduction) require extensive and time-consuming phenotyping, based for example on choice (*Nalam et al., 2018*) or settling (*Annacondia et al., 2021*) assays, and such assays are extremely challenging to perform on large collections. On the other hand, there are increasing numbers of large sequencing datasets, which may also be used to quantify contaminants or microbiome composition (*Sangiovanni et al., 2019*; *Roman-Reyna et al., 2020*; *Gathercole et al., 2021*) and thus as proxies for resistance phenotyping. In this study we investigated such usage of exogenous reads, i.e., reads not mapping to the target reference genome, as a source of information for quantifying herbivore and pathogen abundance in large collections.

We worked with field pennycress (*Thlaspi arvense*), an annual plant in the Brassicaceae family that is increasingly studied as a model species (*Geng et al., 2021*; *Nunn et al., 2022*; *Hu et al., 2022*; *Troyee et al., 2022*; *Galanti et al., 2022*) and new biofuel and winter cover crop (*Dorn et al., 2015*; *Frels et al., 2019*; *Chopra et al., 2019*; *Zhao et al., 2021*). In a previous study, we investigated natural epigenetic variation in a collection of 207 *Thlaspi* lines from across Europe (*Galanti et al., 2022*). Prior to their whole-genome (WGS) and -epigenome sequencing these lines had been grown in a common environment, an open glasshouse where the plants were spontaneously colonized by aphids and powdery mildew, as well as by other microbes. At the time of sequencing, pathogen contamination was still very limited but appeared highly variable, and preliminary analyses showed that it resulted in sizeable amounts of non-target reads assigned to the pest species, i.e., contamination of the DNA samples. Inspired by other recent studies on non-target reads (*Sangiovanni et al., 2019*; *Roman-Reyna et al., 2020*; *Gathercole et al., 2021*), we asked if there was systematic variation in the numbers of aphid and pathogen reads among different *T. arvense* lines, and whether these data, together with our whole-genome plant sequencing data, could provide insights into the genomic basis of plant resistance variation.

The goals of our study were thus twofold: (i) to contribute to a mechanistic understanding of pest resistance in *T. arvense*, and (ii) to explore whether non-target reads from plant sequencing can be used as proxies for studying plant biotic interactions. Considering that we are moving toward an increasingly sequencing-prone world, with more and larger datasets being generated for many species (*Kajiya-Kanegae et al., 2021*; *Colgan et al., 2022*; *Habyarimana et al., 2022*; *Mekbib et al., 2022*; *Metheringham et al., 2022*; *Nocchi et al., 2022*; *Friis et al., 2024*), the use of non-target reads has very broad potential.

## Results

### Reads classification and species identification

Starting from our previously published sequencing data (*Galanti et al., 2022*), the first step of our analysis was to separate the WGS reads of each sample into the ~99.5% mapping to the *T. arvense* reference genome (*Nunn et al., 2022*) and the ~0.5% that did not, hereafter called 'exogenous reads' (*Figure 1A*). Initially, we used all mapped reads for calling variants in *Thlaspi*, but after some difficulties with genome-wide association (GWA) analyses (see below) we suspected that some plant reads were false and mapped to the *T. arvense* genome only because of the high cross-taxa similarity of some genomic regions. We therefore remapped all reads to the genomes of the aphid *Acyrthosiphon pisum*, its endosymbiont *Buchnera aphidicola* and the powdery mildew *Blumeria graminis*, and found that, on average, 7.4% of the reads mapped to both *T. arvense* and at least one of the pests. We removed these ambiguous reads from our analyses and used only the *T. arvense* target reads, 92.1% on average, for variant calling (*Figure 1A*, *Supplementary file 1*).

We next attempted a taxonomic classification of the exogenous reads, in multiple steps. First, we used MG-RAST (*Meyer et al., 2008*; *Keegan et al., 2016*) to assign reads to taxonomic groups based on public sequencing databases. Out of the 78% of the exogenous reads that passed the MG-RAST quality control (*Supplementary file 1*) the majority belonged to bacteria and smaller fractions to fungi, plants, and animals (*Figure 1B* and *Supplementary file 2*). For subsequent group-level analyses, we then focused on nine taxonomic groups that occurred consistently within our samples (*Figure 1C*), and that were particularly abundant or relevant: Erysiphales (fungi), Peronosporales (oomycetes), Aphididae, and Culicidae (both insects), and five bacterial families.

Visual inspection (*Figure 1D*) and other sources of information narrowed down the observed aphid and mildew species to a few candidates. For aphids we considered *A. pisum* (indicated by MG-RAST), *Myzus persicae* (visual match, and a generalist attacking Brassicaceae; *CABI, 2021*) and *Brevicoryne brassicae* (attacks Brassicaceae including *Thlaspi*; *Gabryś and Pawluk, 1999*). For powdery mildew we considered *B. graminis* (indicated by MG-RAST), and *Erysiphe cruciferarum* (attacks Brassicaceae; *Warwick et al., 2002*). To decide among these species, we then used a competitive mapping approach (*Feuerborn et al., 2020*), where the exogenous reads were aligned to a pseudo-reference composed of the same DNA sequences from the different candidate species (see Materials and methods for details, *Supplementary file 3 and 4*). The majority (77%) of the aphid reads mapped to *M. persicae*, 18% to *B. brassicae*, and 5% to *A. pisum*, while 98% of the mildew reads mapped to

**Table 1.** Population differences and SNP-based heritability for different types of exogenous read counts.

Population differences were tested with a linear model, SNP-based heritabilities (and their confidence intervals) estimated with the R package *heritability* (*Kruijer and White, 2019*).

| Taxonomic group | Data type | Population differences ($R^2$ and p-value) | SNP-based heritability |
|---|---|---|---|
| *Myzus persicae* | Mapping to reference genome | 0.245 (p=0.029) | 0.190 (0.055–0.488) |
| *Buchnera aphidicola* | Mapping to reference genome | 0.256 (p=0.016) | 0.169 (0.042–0.490) |
| *Buchnera* | MG-RAST - genus | 0.223 (p=0.090) | 0.115 (0.016–0.505) |
| Aphididae | MG-RAST - family | 0.226 (p=0.082) | 0.189 (0.052–0.496) |
| Culicidae | MG-RAST - family | 0.166 (p=0.519) | 0.183 (0.055–0.465) |
| Erysiphales | MG-RAST - order | 0.326 (p<0.001) | 0.468 (0.238–0.712) |
| Peronosporales | MG-RAST - family | 0.253 (p=0.020) | 0.266 (0.096–0.553) |
| Staphylococcaceae | MG-RAST - family | 0.390 (p<0.001) | 0.301 (0.124–0.567) |
| Burkholderiaceae | MG-RAST - family | 0.275 (p=0.005) | 0.256 (0.092–0.538) |
| Mycobacteriaceae | MG-RAST - family | 0.362 (p<0.001) | 0.294 (0.120–0.560) |
| Pseudomonadaceae | MG-RAST - family | 0.273 (p=0.006) | 0.192 (0.052–0.505) |

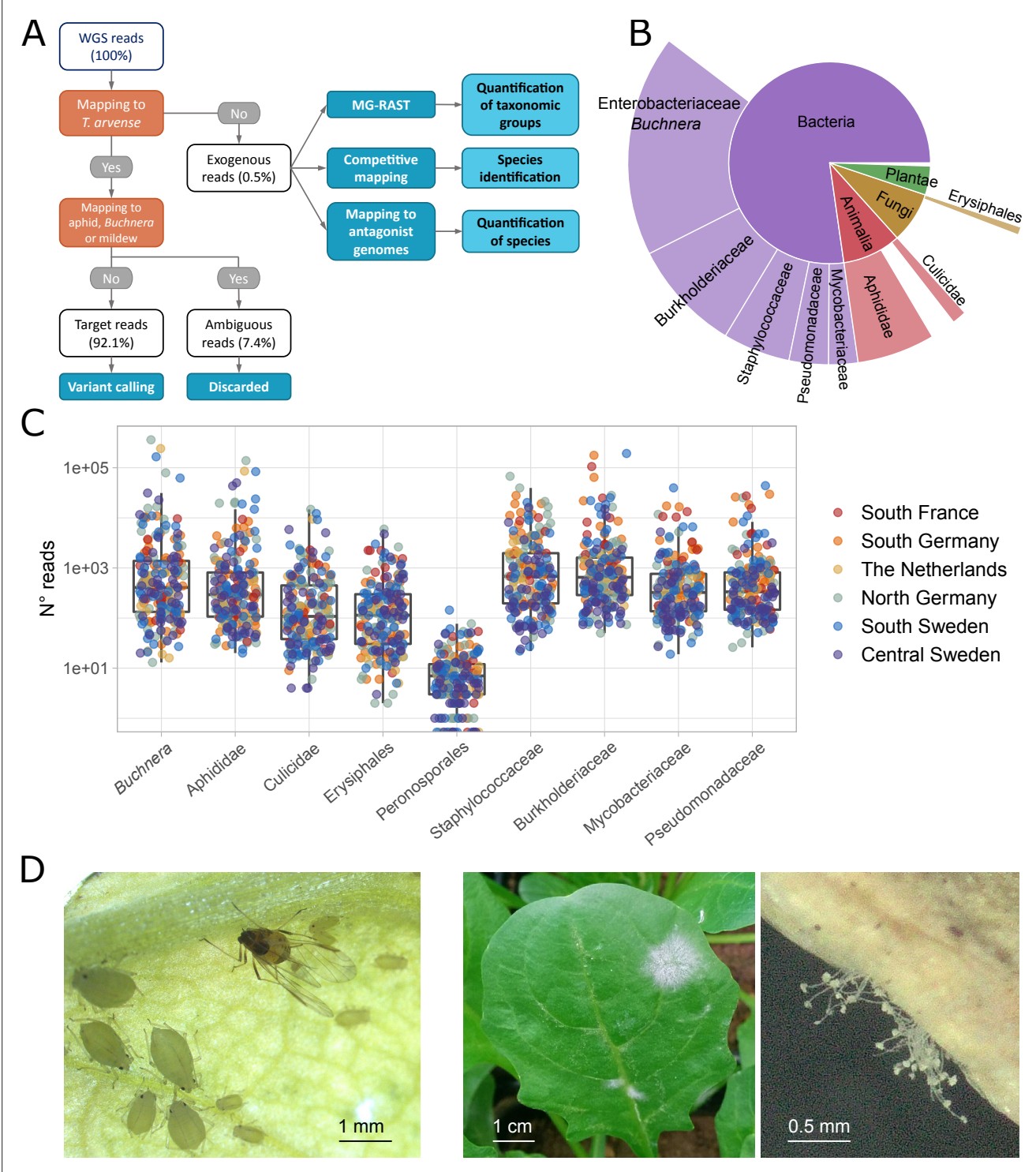

**Figure 1.** Classification of sequencing reads in *T. arvense* whole-genome sequencing (WGS) data. (**A**) Workflow of the analyses, including reads classification (orange nodes) into target, ambiguous, and exogenous reads, and downstream analysis (dark blue nodes) (see Materials and methods). (**B**) Fractions of exogenous reads assigned to different taxonomic groups by MG-RAST (*Meyer et al., 2008*; *Keegan et al., 2016*). (**C**) Read counts assigned to nine selected groups in our 207 *T. arvense* samples from different European regions. (**D**) Aphids and mildew occurring on *T. arvense* leaves during our experiment.

The online version of this article includes the following figure supplement(s) for figure 1:

**Figure supplement 1.** Pest loads in samples with or without pests in the field.

*E. cruciferarum*, with only 2% to the other mildew species (*Supplementary file 5*). Based on these results, we concluded that the plants in our experiment had been attacked by *M. persicae* and *E. cruciferarum*.

Finally, to compare the power of a large database approach (MG-RAST) vs. using specific reference genomes, we also remapped all exogenous reads to the *M. persicae* and *B. aphidicola* genome assemblies (*Singh et al., 2021*) (https://www.ncbi.nlm.nih.gov/datasets/genome/GCF_001939165.1) and used the counts from these two mappings as additional phenotypes, besides the nine taxonomic groups selected through MG-RAST (*Table 1*, *Supplementary file 6*).

## Exogenous read counts are a heritable *Thlaspi* phenotype

As we had observed that aphid and mildew infections in the glasshouse were not random, but prevalent on plants from some origins than others, i.e., possibly reflecting heritable variation in plant resistance, we next tested for population differences and SNP-based heritability in pest and microbiome read counts. Prior to these analyses, to avoid biases caused by different sequencing depths, we corrected the read counts for the total numbers of deduplicated reads in each library and used the residuals as unbiased estimates of aphid, mildew, and microbe loads (see Materials and methods).

For most of the nine taxonomic groups, there were significant population effects, with 20–40% of the variance in read counts explained, as well as significant SNP-based heritability, typically in the range of 0.18–0.30 (*Table 1*). The highest heritability of 0.47 was for read counts of Erysiphales, indicating particularly strong variation for resistance to mildew. Both SNP-based heritability and population differences tended to be stronger for aphid and *Buchnera* data based on read mapping to the reference genomes than for those based on MG-RAST, demonstrating that the former method is more informative and thus preferable if high-quality genome assemblies are available.

An alternative explanation for different aphid and mildew loads in the greenhouse could be that variation in enemy densities in the field was transmitted to the greenhouse, through maternal carry-over effects, or even as seed contamination. However, we had recorded aphid and mildew occurrence during seed sampling in the field and found no significant differences in the glasshouse between the offspring of plants that had been attacked in the field vs. those that had not (*Figure 1—figure supplement 1*).

## Aphid and mildew loads correlate with climate of origin and glucosinolates content of plants

Having established that our method most likely captured variation in plant resistance, we were interested in the ecological drivers of this variation. As climate is known to be a major influence on many biotic interactions as well as plant defenses (*Züst et al., 2012*; *Gao et al., 2019*), we correlated the observed read counts with the climates of origin of the plants. We found negative correlations between aphid read counts and several temperature variables, in particular annual minimum temperature (*Figure 2A*). Aphid read counts were also positively correlated with temperature variability, i.e., the diurnal and annual ranges of temperature (*Figure 2A*). In other words, plants from warmer and more stable climates had consistently lower levels of aphid infestation in our glasshouse, possibly because these plants had evolved greater resistance under such benign climatic conditions where aphids thrive. We found similar, although weaker patterns, for the number of Erysiphales reads. The other analyzed taxonomic groups showed different and often weaker patterns of correlation with climate, except that the read counts of several bacterial groups were positively correlated with annual maximum temperature and in particular diurnal temperature range.

Since glucosinolates are major defense metabolites of Brassicaceae, and their variation could thus be an explanation for variance in plant resistance, we also tested for correlations between the baseline amounts of these metabolites and the frequencies of aphid and mildew reads. Glucosinolate levels were measured on the same *T. arvense* lines in a separate experiment not affected by pests (*Supplementary file 7*). We found positive correlations of aphid read counts with allyl glucosinolate (sinigrin), an aliphatic glucosinolate which is by far the most abundant in the leaves of *T. arvense*, and a stronger negative correlation with benzyl glucosinolates (glucotropaeolin) (*Figure 2B*). Although the baseline levels of benzyl glucosinolates were very low and probably sometimes below the detection level, plant lines where benzyl glucosinolate was detected had significantly lower aphid loads in the glasshouse

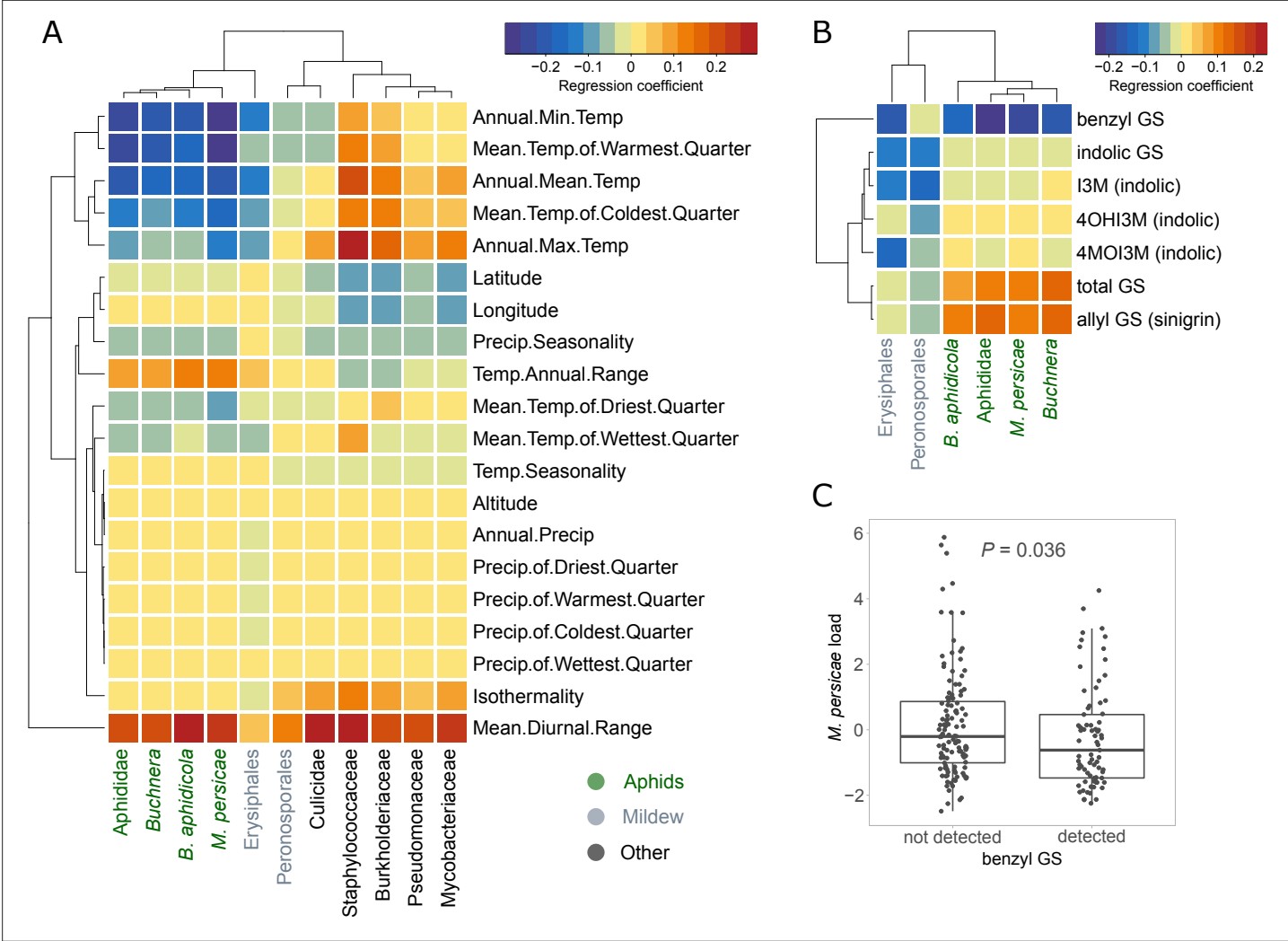

**Figure 2.** Relationships between climates of origin or glucosinolate levels of plants and the exogenous reads loads. (**A**) Correlations with bioclimatic variables. (**B**) Correlations with baseline glucosinolate (GS) levels measured in the same pennycress lines in another experiment. All correlations in (**A**) and (**B**) were done after correction for population structure. Aphid-related read counts are in green, mildew-related in gray, others in black. (**C**) Boxplot of the aphid reads residuals in samples where benzyl GS was detected vs. not. The p-value is based on the Welch's t-test for unequal variances. I3M: indol-3-yl methylGS; 4MOI3M: 4-methoxyindol-3-yl methylGS; 4OHI3M: 1-methoxyindol-3-yl methylGS.

(*Figure 2C*). We also detected three indole glucosinolates, but these did not show any significant correlations with aphid loads.

## GWA identifies peaks near defense genes

To further investigate the genetic basis of variation in aphid, mildew, and microbe loads, we next employed GWA and tested for associations between exogenous read counts and biallelic genetic variants (SNPs and short INDELs). We corrected for population structure using an isolation by state (IBS) matrix and only tested variants with minor allele frequency (MAF)>0.04 (see Materials and methods). Initially, we called genetic variants using all reads that mapped to the *T. arvense* genome and found massive peaks in some highly conserved regions of the genome, which had very high mapping coverage. We suspected that this might be because some non-*Thlaspi* reads were very similar to these highly conserved regions and, by mapping there, generated false variants only in samples containing many non-*Thlaspi* reads. We therefore identified and removed ambiguous reads prior to variant calling, which eliminated the observed massive GWA peaks, indicating that they had indeed reflected false associations (*Figure 3—figure supplement 1*).

After excluding the ambiguous reads, we still found significant GWA peaks for Erysiphales but not for other types of exogenous reads (excluding isolated, unreliable variants) (*Figure 3A*, *Figure 3—figure supplements 2 and 3*). Nevertheless, when clear peaks were visible, regardless of their significance, they were usually located close to genes involved in plant defense response. An enrichment analysis (*Atwell et al., 2010*) confirmed that stronger variants were indeed enriched close to these defense genes (*Supplementary file 8*) for some exogenous read counts (*Figure 3B*, *Figure 3—figure supplements 2 and 3*). For *M. persicae* load there was a peak in the proximity of *Tarvense_01930*, encoding a predicted pathogenesis-related peptide. The top variant in this peak had a slight but clear allelic effect on *M. persicae* load (*Figure 3C*). For Erysiphales load we detected a more persistent enrichment, with a highly significant peak in Scaffold 1, located in a region with several defense genes, including *MAJOR LATEX PROTEINS (MLP)* and two genes similar to *Arabidopsis thaliana SALICYLATE METHYLTRANSFERASE 1 (BSMT1)* (*Figure 3D and E*). This region is wide due to ancient TE colonization, but the top variants are clearly neighboring candidate genes involved in defense (*Figure 3E*). Other significant peaks for Erysiphales load were close to other genes that possibly contribute to resistance such as *PBL7*, involved in signaling and stomatal closure or *SRF3*, reinforcing cell walls by callose deposition.

## Aphid and mildew loads correlate with differential methylation at genes and transposons

Variation in phenotypes, such as our indirect estimates of pest resistance, may not only be associated with DNA sequence but also with epigenetic changes like DNA methylation. This phenotype-associated epigenetic variation can include both heritable and plastic components. The whole-genome bisulfite sequencing (WGBS) data from our previous study (*Galanti et al., 2022*) allowed us to also explore these questions and to test for associations between DNA methylation variation and pest attack. For simplicity, we limited this analysis to *M. persicae* and Erysiphales loads.

Our analysis had two steps: First we called differentially methylated regions (DMRs) between the 20 samples with the most and 20 samples with the least *M. persicae* or Erysiphales loads, and then we conducted epigenome-wide association (EWA) analyses on individual positions located within these DMRs, using the complete dataset (188 lines - see Materials and methods). This approach allowed us to target genomic regions of interest, while strongly reducing the multiple-testing problem of millions of cytosines in the whole genome and correcting for population structure. Using a relaxed false discovery rate (FDR) of 20%, we identified 162 DMRs for *M. persicae* load and 548 DMRs for Erysiphales load (*Figure 4—figure supplement 1*, *Supplementary file 9 and 10*). The majority of these were in the CG context, especially for *M. persicae*-related DMRs (*Figure 4A*, *Figure 4—figure supplement 1*). As observed previously (*Galanti et al., 2022*), DMRs in CHH were generally shorter than in the other sequence contexts (*Figure 4—figure supplement 1*). Since the genome of *T. arvense* is rich in TEs and intergenic regions, the majority of DMRs were located in those features (*Figure 4—figure supplement 1*). However, the DMR density was higher in proximity of genes and particularly in coding sequences (*Figure 4A*), and even DMRs assigned as intergenic (*Figure 4A*) were often located close to genes or promoters. In accordance with previous studies (*Geng et al., 2019*; *Annacondia et al., 2021*), most DMRs were hypomethylated in the infested samples (higher pathogen load), indicating that genes needed for defense might be activated through demethylation.

For a more detailed investigation, we turned to EWA, leveraging the power of the entire *Thlaspi* collection. We tested for associations between *M. persicae* or Erysiphales loads and the methylation at individual cytosines located within the DMRs. As in GWA, we corrected for population structure using an IBS matrix. For both types of pest loads, we found associations in the proximity of genes and especially within TEs, but no genomic feature was particularly enriched for low p-value associations (*Figure 4—figure supplement 2*). *M. persicae* load was associated with methylation at several genomic locations, especially TEs (*Figure 4B*), but these associations had strongly inflated p-values (*Figure 4—figure supplement 2*). For Erysiphales load the p-value distribution was more well-behaved, and we found a clear association with hypomethylation of Copia family 202 TEs upstream of *MAPKK KINASE 20 (MAPKKK20)*, a gene involved in abscisic acid (ABA) stress response and stomatal closure (*Figure 4B, C, and D*). A coverage analysis confirmed that none of the *T. arvense* lines carries insertions or deletions of the TEs upstream of *MAPKKK20*.

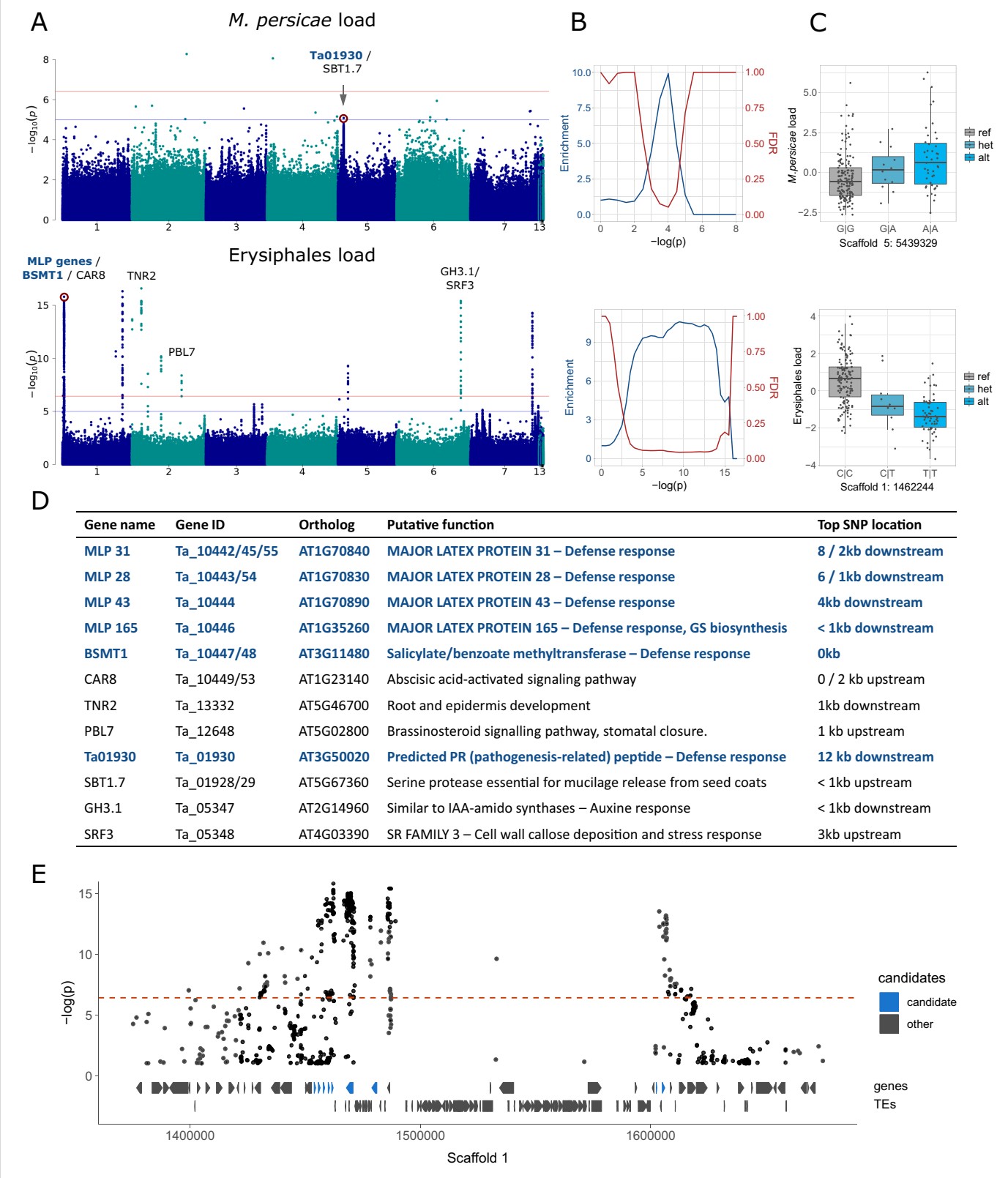

| Gene name | Gene ID | Ortholog | Putative function | Top SNP location |
|---|---|---|---|---|
| **MLP 31** | **Ta_10442/45/55** | **AT1G70840** | **MAJOR LATEX PROTEIN 31 – Defense response** | **8 / 2kb downstream** |
| **MLP 28** | **Ta_10443/54** | **AT1G70830** | **MAJOR LATEX PROTEIN 28 – Defense response** | **6 / 1kb downstream** |
| **MLP 43** | **Ta_10444** | **AT1G70890** | **MAJOR LATEX PROTEIN 43 – Defense response** | **4kb downstream** |
| **MLP 165** | **Ta_10446** | **AT1G35260** | **MAJOR LATEX PROTEIN 165 – Defense response, GS biosynthesis** | **< 1kb downstream** |
| **BSMT1** | **Ta_10447/48** | **AT3G11480** | **Salicylate/benzoate methyltransferase – Defense response** | **0kb** |
| CAR8 | Ta_10449/53 | AT1G23140 | Abscisic acid-activated signaling pathway | 0 / 2 kb upstream |
| TNR2 | Ta_13332 | AT5G46700 | Root and epidermis development | 1kb downstream |
| PBL7 | Ta_12648 | AT5G02800 | Brassinosteroid signalling pathway, stomatal closure. | 1 kb upstream |
| **Ta01930** | **Ta_01930** | **AT3G50020** | **Predicted PR (pathogenesis-related) peptide – Defense response** | **12 kb downstream** |
| SBT1.7 | Ta_01928/29 | AT5G67360 | Serine protease essential for mucilage release from seed coats | < 1kb upstream |
| GH3.1 | Ta_05347 | AT2G14960 | Similar to IAA-amido synthases – Auxine response | < 1kb downstream |
| SRF3 | Ta_05348 | AT4G03390 | SR FAMILY 3 – Cell wall callose deposition and stress response | 3kb upstream |

**Figure 3.** Genome-wide association analyses for aphid and mildew loads. We show only the results for *M. persicae* and MG-RAST Erysiphales read counts; for full results see *Figure 3—figure supplements 2 and 3*. (**A**) Manhattan plots, annotated with genes potentially affecting aphid/mildew colonization. The genome-wide significance (horizontal red line) was calculated based on unlinked variants (*Sobota et al., 2015*), the blue line corresponds to -log(p)=5. (**B**) Corresponding to the Manhattan plots on the left, enrichment of a priori candidates and expected false discovery rates (as

*Figure 3 continued on next page*

*Figure 3 continued*

in **Atwell et al., 2010**) for increasing significance thresholds. (**C**) Allelic effects of the red-marked variants in the corresponding Manhattan plots, with genotypes on the x-axes and the read count residuals on the y-axes. (**D**) The candidate genes marked in panel A, their putative functions and distances to the top variant of the neighboring peak. Candidates in dark blue are the a priori candidates included in the enrichment analyses and involved in defense response (GO:0006952). GS: glucosinolates. (**E**) Zoom-in for the Manhattan plot of Erysiphales load, around the first peak in Scaffold 1, with gene and transposable element (TE) models below, and a priori candidates in blue.

The online version of this article includes the following figure supplement(s) for figure 3:

**Figure supplement 1.** Example of a genome-wide association (GWA) peak caused by ambiguous reads.

**Figure supplement 2.** Genome-wide association (GWA) results for all exogenous reads.

**Figure supplement 3.** Genome-wide association (GWA) results for all exogenous reads.

## Discussion

Plant pests are a major threat to food safety, causing large yield losses, and new crops such as the potential biofuel plant *T. arvense* must be able to resist pathogen and herbivore attacks. A powerful source for obtaining resistant varieties is natural variation in plant defenses, but phenotyping large collections can be very time-consuming and error-prone. Here, we describe how an unplanned pest infestation in a glasshouse experiment, together with available WGS data, can be used to estimate aphid, mildew, and microbial loads, and thus variation in plant resistance. The approach is straightforward, makes use of WGS data without microbiome-specific DNA extraction, and can in principle be applied to many other situations such as field experiments. It is not error-free, but we highlight some potential pitfalls, show how to reduce noise, and illustrate its potential to detect associations with climatic, genetic, and epigenetic variation.

An important first step in our analyses was the identification and classification of pest-related reads in the plant WGS data. We began by classifying all reads as target (only mapping to *T. arvense*), ambiguous (mapping to *T. arvense* and at the same time to at least one of the pest genomes), or exogenous (not mapping to *T. arvense*) (**Figure 1A**). We demonstrated the importance of removing ambiguous reads prior to variant calling, as this prevented calling false positive variants caused by exogenous DNA that also mapped to highly conserved or repetitive sequences in the *T. arvense* genome. We then classified the exogenous reads using MG-RAST (**Meyer et al., 2008**; **Keegan et al., 2016**) or by confident mapping to specific pest genomes, and selected the 11 most relevant and/or abundant taxonomic groups to focus our analyses on. To obtain unbiased pest/microbe loads we also corrected the read counts for the total number of deduplicated reads of each sample. A competitive mapping approach allowed us to identify the aphid and mildew species that had occurred in our experiment as the generalist aphid *M. persicae* and the powdery mildew *E. cruciferarum* (**Figure 2**).

Since we suspected a non-random colonization of pests and microbes in our *T. arvense* collection, we tested for population differences as well as SNP-based heritability. We found significant population differences for most pest and microbe loads, and often heritabilities above 15%, which although low, is still indicative of genetic determination (**Thoen et al., 2017**). Moreover, Erysiphales load had a particularly high heritability of 47% (**Table 1**). We therefore next asked what could explain the observed variation in pest loads in our experiment. As pathogen abundances in the field are often determined by climatic conditions, we expected plants originating from climates less favorable to aphids to perform worse in our glasshouse, i.e., to have higher pathogen loads. As expected, aphid counts were negatively correlated with temperature of origin (particularly minimum temperature), and positively with temperature variability (mean diurnal range and temperature annual range) (**Figure 3A**), suggesting that plants from colder and more thermally fluctuating climates, which are less favorable to aphids, were less well defended and performed worse in our glasshouse. We found similar but weaker patterns for Erysiphales load.

As we expected the observed climate-associated variation in pest loads to be at least partially explained by variation in chemical defenses, and since in *A. thaliana* glucosinolates, the main defensive compounds, are known to be geographically structured in response to aphid distributions (**Züst et al., 2012**), we also tested for association of aphid and mildew loads with glucosinolates in our collection. In accordance with literature on *A. thaliana* (**Kim and Jander, 2007**), we observed a positive correlation of aphid loads with total glucosinolates as well as with the most abundant glucosinolate sinigrin (aliphatic glucosinolate), but a negative correlation with benzyl glucosinolate (**Figure 3B**).

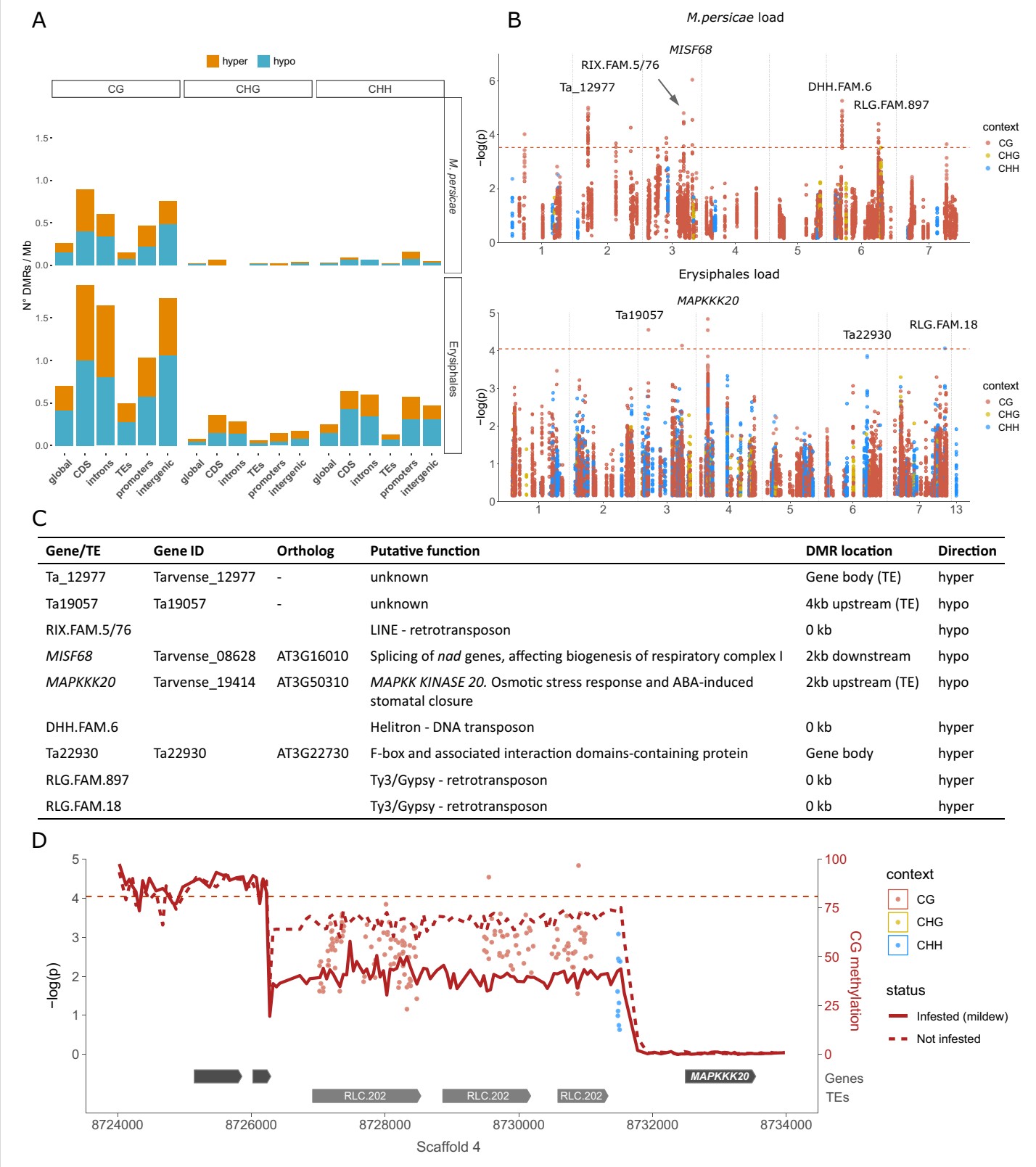

**Figure 4.** Differential methylation associated with aphid and mildew loads. (**A**) Differentially methylated region (DMR) densities in different genomic features when comparing the 20 samples with the most vs. the least *M. persicae* (top) or Erysiphales (bottom) load. CDS: coding sequences. (**B**) Manhattan plots from epigenome-wide association (EWA) analyses based on individual cytosines within DMRs, with sequence contexts in different colors and annotation of genes close to low p-value cytosines. The genome-wide Bonferroni significance thresholds (dashed red lines) were calculated

*Figure 4 continued on next page*

*Figure 4 continued*

based on the number of DMRs. (**C**) Candidate genes and transposable elements (TEs) marked in panel B, their putative functions, genomic locations of associated DMRs, and whether infested samples were hyper- or hypomethylated. (**D**) Zoomed-in Manhattan plot for Erysiphales load around the peak in Scaffold 4, with gene and TE models given below. The CG methylation in the 20 most and least infested samples was calculated over 50 bp bins (see *Figure 4—figure supplement 2* for methylation in other contexts).

The online version of this article includes the following figure supplement(s) for figure 4:

**Figure supplement 1.** Lengths and genomic locations of differentially methylated regions (DMRs).

**Figure supplement 2.** Epigenome-wide association (EWA) enrichment in different genomic features and p-value distributions.

These findings suggest that glucosinolate composition, rather than total amount, is important for aphid defense, and that while benzyl glucosinolate might have a deterrent effect, sinigrin might on the contrary attract *M. persicae* or act as a stimulant, which would be in accordance with previous observations (*Klingauf et al., 1972*).

To detect genetic variants associated with pest and microbe loads, we then conducted a GWA study (GWAS). For aphid (*M. persicae*) load, we detected only one non-significant peak in Scaffold 5, close to a pathogenesis-related coding gene (*Figure 3A and D*). For Erysiphales load, however, there were several significant associations neighboring genes directly involved in defense, mostly members of the *MLP* family, clustered in a large peak on the first arm of Scaffold 1 (*Figure 3E*). *MLP165*, the closest gene to the most significant variant in the peak, is indirectly involved in glucosinolate biosynthesis in *A. thaliana* (*The Arabidopsis Information Resource (TAIR), 2000*), which might explain why baseline glucosinolate levels were associated with Erysiphales load (*Figure 3B*). Further GWA peaks for Erysiphales pointed toward other genes indirectly involved in the defense response through phytohormone signaling (e.g. *CAR8, PBL7, GH3.1*) or preventing pathogen access through cell wall reinforcement or stomatal closure (*SRF3, PBL7*) (*Figure 3D*). Further experiments would be necessary to confirm the functionality of these genes.

An important general insight from our GWA analyses was the frequent ambiguity of reads that mapped to both pest and host plant genomes. Such ambiguous reads generated false variants only present in samples with pest DNA, which resulted in highly significant false associations, and it was therefore important to remove these reads before variant calling. Another potential reason for sequence similarity between host and pathogens could be defense mechanisms such as RNA interference. If *T. arvense* produces small or micro-RNAs to silence pathogen genes, this would originate from genomic regions of high similarity between the host and the pathogen, and thus reflect a true association. However, a BLAST (*Camacho et al., 2009*) of the region in which the suspicious associations occurred did not reveal any similarity to aphid or mildew genes, but instead to the highly conserved ribosomal RNA coding regions. While genetic variants are generally inherited from parents and thus reflect evolutionary processes, DNA methylation variants can be heritable but can also reflect plastic responses to environmental stresses like herbivores or pathogens. Our data do not allow to confidently distinguish between these two sources of DNA methylation variation, and thus should be interpreted with caution, especially with regard to the directionality of associations. A beneficial DNA methylation variant is expected to be associated with lower pathogen load when already present before pathogen arrival, but with higher pathogen load when plastically induced by pathogens during the experiment. For both *M. persicae* and Erysiphales, the majority of DMRs were hypomethylated in affected samples, which is in accordance with the loss of methylation observed in *A. thaliana* and *T. arvense* upon aphid feeding, and in diploid wheat upon powdery mildew infection (*Geng et al., 2019*; *Annacondia et al., 2021*; *Troyee et al., 2022*), but we also detected hypermethylation at several loci. *M. persicae* load was associated with differential methylation at only few genes but several TEs, which is in accordance with the aphid or stress-induced TE reactivation observed in *A. thaliana* (*Annacondia et al., 2021*; *Roquis et al., 2021*). Erysiphales load was associated with hypomethylated Copia TEs upstream of *MAPKKK20*, a gene involved in ABA-mediated signaling and stomatal closure. Since stomatal closure is a known defense mechanism to block pathogen access (*Melotto et al., 2017*), it is tempting to conclude that hypomethylation of the *MAPKKK20* promoter might induce its overexpression and consequent stomatal closure, thereby preventing mildew access to the leaf blade. Overall, we found associations between pathogen load and TE methylation that could potentially act both in *cis* (e.g. Copia TE methylation in *MAPKKK20* promoter) and in *trans*, e.g., through transposon reactivation (e.g. LINE, Helitron, and Ty3/Gypsi

TEs isolated from genes). Although we cannot confidently distinguish inherited vs. induced DNA methylation variants, to our knowledge this is the first coupled GWA-EWA analysis conducted on a large natural plant collection.

In summary, our study offers first insights into the defense mechanisms of *T. arvense*, including candidate genes and alleles which may be of interest for breeding efforts in this novel biofuel and cover crop. It also provides a proof of principle that exogenous reads from large sequencing efforts, usually discarded if not mapping to the target genome, can be leveraged to extract additional information about important biotic interactions of the target species, including its antagonists and microbiome components. We combined this approach with data from a common environment experiment to show that pest and microbiome load were geographically structured, as expected from locally adapted traits, and associated with both genetic and DNA methylation variants. In principle, our approach can be applied to many other designs. For example, field-collected samples could be used to quantify geographic pathogen distributions. With the decreasing cost of sequencing and increasing large-scale and single-species sequencing projects (e.g. *Kajiya-Kanegae et al., 2021*; *Colgan et al., 2022*; *Habyarimana et al., 2022*; *Mekbib et al., 2022*; *Metheringham et al., 2022*; *Nocchi et al., 2022*; *Friis et al., 2024*), the number of datasets suitable for such analyses is set to rapidly increase in the near future.

## Materials and methods
### Plant growth and sequencing
The WGS data used in this study were already published in *Galanti et al., 2022*. Please refer to this publication for details on data generation and methods. Briefly, we collected 207 *T. arvense* accessions from 36 European populations in July 2018, and we grew their offspring in a completely randomized design, in a glasshouse at the University of Tübingen (48°32'21.3"N, 9°02'04.2"E) between August and October 2019. The glasshouse was located in biodiverse surroundings, and insects and pests could enter when the windows opened for temperature regulation. A few weeks after germination, we noticed aphid and mildew infestations. After 46 days we sampled the third or fourth true leaf of each plant and snap-froze it in liquid nitrogen. We extracted DNA using the DNeasy Plant Mini Kit (QIAGEN, Hilden, DE), sonicated (Covaris) 300 ng of genomic DNA and used the NEBNext Ultra II DNA Library Prep Kit for Illumina (New England Biolabs) to prepare the libraries. Half way through the protocol we split the DNA into 1/3 for genomic libraries and 2/3 for bisulfite libraries. For the bisulfite conversion we used the EZ-96 DNA Methylation-Gold MagPrep (ZYMO) kit. We sequenced paired-end for 150 cycles using Illumina NovaSeq 6000 (Illumina, San Diego, CA, USA) for genomic libraries and HiSeq X Ten (Illumina, San Diego, CA, USA) for bisulfite libraries.

### Reads mapping and classification
Upon demultiplexing the raw reads, we used cutadapt (*Martin, 2011*) for quality (minimum quality of 20) and adaptor trimming, excluding reads shorter than 35 bp. We used FastQC and MultiQC (*Andrews, 2010*; *Ewels et al., 2016*) to estimate the duplication rate, and calculated the total deduplicated reads, which we later used for correcting the number of exogenous reads. We then classified the reads based on their mapping behavior. First we aligned reads to the *T. arvense* reference genome (*Nunn et al., 2022*) with BWA-MEM v0.7.17 (*Li and Durbin, 2009*), excluding multimapping reads (-c 1) and marking duplicates with MarkDuplicatesSpark (*Van der Auwera et al., 2013*; *Poplin et al., 2018*). We then mapped all samples again (*Li and Durbin, 2009*) to the three putative exogenous genomes of pea aphid (*Acrophyson pisum*), the aphid symbiont *B. aphidicola* and powdery mildew (*B. graminis*), using available resources (https://www.ncbi.nlm.nih.gov/assembly/GCF_005508785.2, https://www.ncbi.nlm.nih.gov/datasets/genome/GCF_001939165.1) (*Frantzeskakis et al., 2018*). After this, we used a custom script to collect all read IDs within a sample mapping to any of the three exogenous genomes, and removed any of these reads from the *T. arvense* alignment bam files. We thus removed all ambiguous reads before proceeding with variant calling. To compare coverage of specific regions with and without ambiguous reads, we used samtools bedcov (*Danecek et al., 2021*). The numbers of reads classified by their mapping behavior are reported in *Supplementary file 1*.

## Variant calling

For variant calling we used GATK4 v4.1.8.1 (*Van der Auwera et al., 2013*; *Poplin et al., 2018*), following the best practices for germline short variant discovery (https://gatk.broadinstitute.org/hc/en-us/articles/360035535932-Germline-short-variant-discovery-SNPs-Indels-) with few adjustments for large datasets (*Galanti et al., 2022*). Briefly, starting from the bam files generated after the removal of ambiguous reads, we (i) ran HaplotypeCaller, (ii) combined the resulting GVCF files with GenomicsDBImport and GenotypeGVCFs, and (iii) filtered out low-quality variants with VariantFiltration (see *Galanti et al., 2022* for more details). Finally, we used vcftools v0.1.16 (*Danecek et al., 2011*) to retain biallelic variants with MAF>0.01 and a maximum of 10% missing genotype calls. We imputed these missing calls with BEAGLE 5.1 (*Browning et al., 2018*) to obtain a complete multisample vcf file.

## Identification and classification of exogenous reads

To identify exogenous reads, we extracted all unmapped reads from the bam files created aligning WGS reads to the *T. arvense* genome (*Nunn et al., 2022*). We selected reads with both mates unmapped (SAM flag 12) and excluded supplementary alignments (SAM flag 256 after running MarkDuplicatesSpark) with samtools (*Danecek et al., 2021*). We then recovered these reads from the trimmed fastq files with seqtk subset (https://github.com/lh3/seqtk; *Li, 2024*) to obtain fastq files of unmapped reads only. We used these as input for MG-RAST (*Meyer et al., 2008*; *Keegan et al., 2016*), a web-based tool for phylogenetic analysis of metagenomes.

We ran MG-RAST mostly with default parameters, without assembled reads, excluding dereplicated sequences, and dynamically trimming reads with a minimum Phred score of 15 in more than 5 consecutive bases. We set the 'sequence screening' to *A. thaliana*, the closest relative of *T. arvense* available. We used two different approaches to extract read counts. First, we classified all reads up to family level using the web-based Analysis tool from MG-RAST. We used RefSeq as query annotation database and filtered reads classified with low confidence using default settings: e-value 5, 60 %-identity, length 15, and min.abundance of 1 (*Supplementary file 2*). Out of the hundreds of taxonomic groups identified by MG-RAST, we selected only a small subset for follow-up analyses, based on their biological relevance, our visual observations and/or abundance: Aphididae, Culicidae, Peronosporales, Staphylocaccaceae, Burkholderiaceae, Mycobacteriaceae, and Pseudomonadaceae (*Table 1*). Additionally, we used a custom Python script to download individual 'taxonomy' or 'sequence_breakdown' results from MG-RAST API (*Paczian et al., 2019*) and extracted the counts of the genus *Buchnera,* including bacterial symbionts of many aphid species, and of the order Erysiphales, to quantify the observed mildew infection (*Table 1*). All the code for extracting counts for all families or specific taxonomic groups are available on GitHub (https://github.com/junhee-jung/MG-RAST-read-counter, copy archived at *Jung and Galanti, 2024*).

In addition to the nine read groups selected from MG-RAST results, we also performed a highly confident mapping of exogenous reads to the *M. persicae* and *B. aphidicola* genome assemblies (*Singh et al., 2021*) (https://www.ncbi.nlm.nih.gov/datasets/genome/GCF_001939165.1), to test whether mapping to a high-quality assembly of the exact pathogen has a higher sensitivity than MG-RAST. We mapped with BWA-MEM v0.7.17 (*Li and Durbin, 2009*), using a seed length of 25 bp (*Robinson et al., 2017*) and removing reads with MAPQ<20 and duplicates with MarkDuplicatesSpark (*Van der Auwera et al., 2013*; *Poplin et al., 2018*). We then counted all reads in the bam files.

Finally, we log transformed all read counts to approximate normality, and corrected for the total number of deduplicated reads by extracting residuals from the following linear model, log(read_count +1)~log(deduplicated_reads), which allowed us to quantify non-*Thlaspi* loads, correcting for the sequencing depth of each sample.

## Exogenous reads heritability and species identification

To exclude the possibility that aphid and mildew infestation patterns were carried over from the field, through seed contamination or maternal effects, we used aphid and mildew presence/absence data collected in the field. We found no difference in aphid or mildew loads between samples with and without aphids or mildew on the original parental plant in the field (*Figure 1—figure supplement 1*). Nevertheless, to exclude a possible bias, we excluded one outlier sample with particularly high aphid load and aphids observed in the field (*Figure 1—figure supplement 1*) from the analyses.

To test for variation between populations we used a general linear model with population as a predictor. To measure SNP-based heritability, i.e., the proportion of variance explained by kinship, we used the marker_h2() function from the R package *heritability* (*Kruijer and White, 2019*), which uses a genetic distance matrix as predictor to compute REML-estimates of the genetic and residual variance. We used the same IBS matrix as for GWAS and for the correlations with climatic variables.

Even though MG-RAST classifies reads based on all taxonomic ranks, the accuracy of species identification of course strongly depends on the sequences available in the query databases. MG-RAST assigned our aphid reads to *A. pisum*, but this did not fit with our visual observations and with the poor performance of this species on Brassicaceae (*Prince et al., 2014*). We therefore selected three plausible aphid species and test which of these had mostly likely attacked our experiment. In addition to *A. pisum*, we tested two other aphid species commonly attacking Brassicaceae: *B. brassicae* and *M. persicae*. While not all three species have reference genomes available, all mitochondrial genomes are available on NCBI (*NCBI, 1988*) under accession numbers MN232006, NC_011594, and NC_056270. We downloaded these sequences, aligned them to each other (*Sievers et al., 2011*), removed INDELs to retain only SNPs, and combined them into a single pseudo-reference fasta file (*Supplementary file 3*). We then mapped the exogenous reads from 40 randomly selected samples to this pseudo-reference, allowing for unique mappings only and counted the reads mapping to either of the three aphid species. We used the same approach for mildew except that we included only two possible species: *B. graminis*, as suggested by MG-RAST, and *E. cruciferarum* which is known to attack Brassicaceae but was not in the MG-RAST query database and seemed plausible from visual inspection (*Figure 2B*). For the mildew pseudo-reference (*Supplementary file 4*) we used the internal transcribed spacer, which is publicly available for both species on NCBI (*NCBI, 1988*) under accession numbers MT644878 and AF031283.

## Quantification of glucosinolates

Using seed material collected from the sequenced plants, we conducted a follow-up experiment to estimate the glucosinolate contents of all 207 lines in the absence of pathogens. Briefly, we sowed the seeds in Petri dishes, stratified them at 4°C in the dark for 2 weeks and transplanted the germinated seedlings to individual 9×9 cm$^2$ pots. We grew the plants in a growth chamber with a 14/10 hr light/dark cycle at 21/17°C and a relative humidity of ~45%. Two weeks after germination the plants were vernalized at 4°C for 2 more weeks in order to minimize phenological and developmental differences between winter and summer annuals. Ten days after vernalization, we collected the third or fourth true leaf and snap-froze it in liquid nitrogen. After freeze-drying, we weighed all samples and extracted the material threefold in 80% methanol, adding *p*-hydroxybenzl glucosinolate (Phytoplan, Heidelberg, Germany) as internal standard. After centrifugation, we applied the supernatants onto ion-exchange columns with diethylaminoethyl Sephadex A25 (Sigma-Aldrich, St. Louis, MO, USA) in 0.5 M acetic acid buffer, pH 5. We added purified sulfatase, converting glucosinolates to desulfo glucosinolates. After 1 day, we eluted desulfo glucosinolates in water and analyzed them on a HPLC coupled to a DAD detector (HPLC-1200 Series, Agilent Technologies, Inc, Santa Clara, CA, USA) equipped with a Supelcosil LC 18 column (3 μm, 150×3 mm, Supelco, Bellefonte, PA, USA). We analyzed the samples with a gradient from water to methanol starting at 5% methanol for 6 min and then increased from 5% to 95% within 13 min with a hold at 95% for 2 min, followed by a column equilibration cycle. We identified different glucosinolates based on their retention times and UV spectra in comparison to respective standards and an in-house database. We integrated peaks at 229 nm and calculated respective glucosinolate concentrations in relation to the internal standard and sample dry mass, using response factors as reported by *Agerbirk et al., 2015*.

## Drivers of exogenous reads variation

To test for associations between glucosinolate variation, as well as climate of origin, and the observed pest loads, we extracted average bioclimatic variables for the 25 years predating our experiment for our 36 study populations from the Copernicus website (*ECMWF, 2020*), as described in *Galanti et al., 2022*. We then used the R package *lme4qtl* (*Ziyatdinov et al., 2018*) to run mixed models that included either bioclimatic variables or glucosinolate contents as explanatory variables, and the exogenous read counts as dependent variables, while correcting for population structure with the same IBS matrix as in GWA and EWA analyses (see below).

## GWA analysis

We conducted GWA with mixed models that corrected for population structure with a genetic IBS matrix as a random factor, as implemented in GEMMA (*Zhou and Stephens, 2012*). To obtain the IBS matrix we used PLINK v1.90b6.12 (*Purcell et al., 2007*). Starting from the imputed multisample vcf file obtained from variant calling, we pruned variants with LD>0.8 in 50 variants windows, sliding by five. To produce the genetic variants used for GWAS, we also started from the imputed multisample vcf file from variant calling and filtered out variants with MAF<0.04. As phenotypes we used the number of exogenous reads corrected for the total number of deduplicated reads, as described above.

To validate our results and test for overlap with existing gene functional annotations, we performed an enrichment analysis of variants neighboring a priori candidate genes as described in *Atwell et al., 2010*. Briefly, we attributed a priori candidate status to all variants located within 20 kb from orthologs (*Emms and Kelly, 2019*) of *A. thaliana* genes annotated with the GO term 'defense response' (GO:0006952), including nine genes similar to AtBSMT1 (*Supplementary file 8*). We then calculated enrichment for incremental -log(p) thresholds as the ratio between observed frequency (significant a priori candidate/significant variants) and background frequency (total a priori candidate/total variants), and an upper bound for the FDR (*Galanti et al., 2022*; *Atwell et al., 2010*). We further assessed the significance of the enrichment through a previously established genome rotation scheme (*Nordborg et al., 2005*). Briefly, we calculated a null distribution of enrichments by randomly rotating the p-values and a priori candidate status of the genetic variants within each chromosome for 1 M permutations. We then assessed significance by comparing the observed enrichment at the Bonferroni threshold to the null distribution. The code for these analyses is available on https://github.com/Dario-Galanti/multipheno_GWAS/tree/main/gemmaGWAS (copy archived at *Galanti, 2024d*).

## Methylation and DMR calling

For the methylome analyses we used the EpiDiverse toolkit (*Nunn et al., 2021*), specifically designed for large WGBS datasets. We used the WGBS pipeline (https://github.com/EpiDiverse/wgbs; *Nunn, 2022*) for read mapping and methylation calling, retained only uniquely mapping reads longer than 30 bp, and obtained individual-sample bedGraph files for each sequence context. We then called DMRs using the DMR pipeline (*Nunn et al., 2021*), with a minimum coverage of 4×. We compared the 20 samples with the most and the least *M. persicae* and Eriysiphales loads, resulting in two sets of DMRs for each sequence context. Since this was only the first step of our methylation analysis, meant to identify potential regions of interest, we retained all DMRs with an FDR<20%. To understand the genomic preferences of DMRs, we intersected them with genomic features and calculated their densities in each by dividing their number by the total Mb covered by each genomic feature in the genome.

## EWA analysis

Following the DMR calling, we investigated methylation-phenotype relationships in more detail, using EWA. We ran EWA similarly to GWA, enabling the '-notsnp' option available in GEMMA (*Zhou and Stephens, 2012*), and correcting for population structure with the same IBS matrix. To exclude possible biases, we excluded all samples with a bisulfite non-conversion rate >1 (*Galanti et al., 2022*), which left 188 samples for analysis. To generate the methylation input files we first used custom scripts (https://github.com/Dario-Galanti/WGBS_downstream/tree/main/WGBS_simpleworkflow, copy archived at *Galanti, 2024c*; *Galanti et al., 2022*) to unite individual-sample bedGraph files into unionbed files and retain positions with coverage>3 in at least 95% of the samples and a methylation difference of at least 5% in at least two samples. We then intersected the unionbed files with the DMRs of the corresponding sequence context using bedtools (*Quinlan and Hall, 2010*) and converted unionbed to BIMBAM format as input for GEMMA.

We ran EWA for individual positions within the DMRs and calculated Bonferroni thresholds based on the number of DMRs, assuming that cytosines within the same DMR are mostly autocorrelated. To observe in which genomic features associations with lower p-values were located, we performed enrichment analyses similar to the ones performed for defense a priori candidate genes in GWA (*Atwell et al., 2010*), but based on whole genomic features. Starting from all cytosines used for EWA, we calculated the background frequency as the fraction of all cytosines located in each genomic feature and then calculated the observed frequency in the same way for -log(p) 0.5 increments, with enrichment as the ratio of observed and expected frequencies. All code used for EWA and the

enrichment analysis in genomic features is available on https://github.com/Dario-Galanti/EWAS/tree/main/gemmaEWAS (copy archived at *Galanti, 2024e*).

## Acknowledgements

We thank the EpiDiverse network for its amazing support and discussions, in particular Adrián Contreras-Garrido, Bárbara Díez Rodríguez, Iris Sammarco, Adam Nunn, Daniela Ramos, and Anupoma Troyee for their close collaboration. We also thank Frank Reis for his feedback and expert tips for the project, Cecilia Heyworth for proofreading the manuscript and Karin Djendouci at Bielefeld University for support in glucosinolate analysis. We thank Peter Stadler at the University of Leipzig and David Langenberger from ecSeq, which helped with computing and hosted the EpiDiverse servers. The BinAC cluster is managed by the High Performance and Cloud Computing Group at the Zentrum für Datenverarbeitung of the University of Tübingen, and supported by the state of Baden-Württemberg through bwHPC and the German Research Foundation (DFG) through grant no INST 37/935-1 FUGG.

## Additional information

### Funding

| Funder | Grant reference number | Author |
| --- | --- | --- |
| Horizon 2020 Framework Programme | 764965 | Oliver Bossdorf |
| Deutsche Forschungsgemeinschaft | 401829393 | Oliver Bossdorf |

The funders had no role in study design, data collection and interpretation, or the decision to submit the work for publication.

### Author contributions

Dario Galanti, Conceptualization, Data curation, Formal analysis, Investigation, Visualization, Methodology, Writing – original draft, Writing – review and editing; Jun Hee Jung, Conceptualization, Data curation, Methodology, Writing – review and editing; Caroline Müller, Investigation, Writing – review and editing; Oliver Bossdorf, Conceptualization, Supervision, Funding acquisition, Methodology, Writing – original draft, Writing – review and editing

### Author ORCIDs

Dario Galanti ⓘ https://orcid.org/0000-0002-6567-1505
Jun Hee Jung ⓘ https://orcid.org/0000-0002-6523-4794
Caroline Müller ⓘ https://orcid.org/0000-0002-8447-534X
Oliver Bossdorf ⓘ https://orcid.org/0000-0001-7504-6511

Reviewer #1 (Public review): https://doi.org/10.7554/eLife.95510.3.sa1
Reviewer #2 (Public review): https://doi.org/10.7554/eLife.95510.3.sa2
Author response https://doi.org/10.7554/eLife.95510.3.sa3

## Additional files

### Supplementary files

• Supplementary file 1. Classification of reads. Total reads, duplication rates, deduplicated raw reads, target, ambiguous, and exogenous reads and MG-RAST reads passing QC.

• Supplementary file 2. MG-RAST classification of exogenous reads. Results of MG-RAST classification of exogenous reads at the family level, using RefSeq as query database.

• Supplementary file 3. Competitive mapping pseudo-reference for aphid identification. Fasta file combining mitochondrial sequences of *B. Brassicaceae, A. pisum,* and *M. persicae*, with structural variants removed.

- Supplementary file 4. Competitive mapping pseudo-reference for mildew identification. Fasta file combining internal transcribed spacer (ITS) sequences of *B. graminis* and *E. cruciferarum*, with structural variants removed.

- Supplementary file 5. Results of competitive mapping for pest identification. Number of reads mapping uniquely to the pseudo-reference genomes of different aphid (mitochondrial DNA for either *B. Brassicaceae*, *A. pisum*, or *M. persicae*) or mildew species (internal transcribed spacer [ITS] sequence for either *B. graminis* or *E. cruciferarum*).

- Supplementary file 6. Exogenous reads used for downstream analyses. Classes of exogenous reads used in the analyses, including the nine groups from MG-RAST and two from mapping to the *M. persicae* and *B. aphidicola* reference genomes.

- Supplementary file 7. Quantification of glucosinolates. Glucosinolate concentrations in leaves of *T. arvense*, obtained from offspring of the sequenced plants, not infested by any herbivores nor pathogens.

- Supplementary file 8. A priori 'defense response' candidate genes used for the genome-wide association (GWA) enrichment analysis. List of *T. arvense* candidate genes used for the GWA enrichment analysis: orthologs of *A. thaliana* genes annotated with the GO term 'defense response'.

- Supplementary file 9. Differentially methylated regions (DMRs) based on *M. persicae* load. List of DMRs called between the 20 samples with the highest and the lowest *M. persicae* load.

- Supplementary file 10. Differentially methylated regions (DMRs) based on Erysiphales load. List of DMRs called between the 20 samples with the highest and the lowest Erysiphales load.

- MDAR checklist

## Data availability

The seed material from the sequenced lines is available at the Nottingham Arabidopsis Stock Centre (NASC) under stock numbers N950001 to 950204. Genomic and bisulfite sequencing raw data are available on the ENA Sequence Read Archive (https://www.ebi.ac.uk/ena) under study accession number PRJEB50950. The reference genome and annotations were previously published by *Nunn et al., 2022*. GWA and EWA results in a format compatible with the Integrative Genomics Viewer (https://www.igv.org/) are available on Zenodo (https://zenodo.org/records/10011535). All code used in this study is available and documented on GitHub. The scripts for variant calling, filtering and imputation are on https://github.com/Dario-Galanti/BinAC_varcalling (copy archived at *Galanti, 2024a*), and the scripts for the classification of sequencing reads and MG-RAST analysis are in https://github.com/Dario-Galanti/Exoreads_treasure and https://github.com/junhee-jung/MG-RAST-read-counter respectively (copies archived at *Galanti, 2024b* and *Jung and Galanti, 2024*). The pipelines for methylation and DMR calling from WGBS data can be found on the EpiDiverse GitHub (https://github.com/EpiDiverse). The workflow for downstream analysis of methylation data is on https://github.com/Dario-Galanti/WGBS_downstream/tree/main/WGBS_simpleworkflow (copy archived at *Galanti, 2024c*). Finally, the scripts for running GWA and EWA analysis are on https://github.com/Dario-Galanti/multipheno_GWAS/tree/main/gemmaGWAS and https://github.com/Dario-Galanti/EWAS/tree/main/gemmaEWAS respectively (copies archived at *Galanti, 2024d* and *Galanti, 2024e*).

The following dataset was generated:

| Author(s) | Year | Dataset title | Dataset URL | Database and Identifier |
|---|---|---|---|---|
| Galanti D, Jung JH, Müller C, Bossdorf O | 2023 | Discarded sequencing reads uncover natural variation in pest resistance in Thlaspi arvense | https://doi.org/10.5281/zenodo.10011535 | Zenodo, 10.5281/zenodo.10011535 |

The following previously published dataset was used:

| Author(s) | Year | Dataset title | Dataset URL | Database and Identifier |
|---|---|---|---|---|
| Galanti D, Ramos-Cruz D, Nunn A, Rodríguez-Arévalo I, Scheepens JF, Becker C, Bossdorf O | 2022 | Genetic and environmental drivers of large-scale epigenetic variation in Thlaspi arvense | https://www.ebi.ac.uk/ena/browser/view/PRJEB50950 | European Nucleotide Archive, PRJEB50950 |

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
