## [Editor Report · eLife Assessment]

This **important** study presents a significant methodological advance by leveraging previously discarded, unmapped DNA sequence reads to estimate pest infestation loads across plant accessions, and map variation in these apparent pest loads to defense genes. The bioinformatics approach is **compelling**, and the results should bear broad implications for phenotype-genotype prediction, especially regarding the use of unmapped reads for GWAS.

---

## [Referee Report · Reviewer #1 (Public review)]

Galanti et al. present an innovative new method to determine the susceptibility of large collections of plant accessions towards infestations by herbivores and pathogens. This work resulted from an unplanned infestation of plants in a greenhouse that was later harvested for sequencing. When these plants were extracted for DNA, associated pest DNA was extracted and sequenced as well. In a standard analysis, all sequencing reads would be mapped to the plant reference genome and unmapped reads, most likely originating from 'exogenous' pest DNA, would be discarded. Here, the authors argue that these unmapped reads contain valuable information and can be used to quantify plant infestation loads.

For the present manuscript, the authors re-analysed a published dataset of 207 sequenced accessions of Thlaspi arvense. In this data, 0.5% of all reads had been classified as exogenous reads, while 99.5% mapped to the T. arvense reference genome. In a first step, however, the authors repeated read mapping against other reference genomes of potential pest species and found that a substantial fraction of 'ambiguous' reads mapped to at least one such species. Removing these reads improved the results of downstream GWAs, and is in itself an interesting tool that should be adopted more widely.

The exogenous reads were primarily mapped to the genomes of the aphid Myzus persicae and the powdery mildew Erysiphe cruciferarum, from which the authors concluded that these were the likely pests present in their greenhouse. The authors then used these mapped pest read counts as an approximate measure of infestation load and performed GWA studies to identify plant gene regions across the T. arvense accessions that were associated with higher or lower pest read counts. In principle, this is an exciting approach that extracts useful information from 'junk' reads that are usually discarded. The results seem to support the authors' arguments, with relatively high heritabilities of pest read counts among T. arvense accessions, and GWA peaks close to known defence genes. Nonetheless, I do feel that more validation would be needed to support these conclusions, and given the radical novelty of this approach, additional experiments should be performed.

A weakness of this study is that no actual aphid or mildew infestations of plants were recorded by the authors. They only mention that they anecdotally observed differences in infestations among accessions. As systematic quantification is no longer possible in retrospect, a smaller experiment could be performed in which a few accessions are infested with different quantities of aphids and/or mildew, followed by sequencing and pest read mapping. Such an approach would have the added benefit of allowing causally linking pest read count and pest load, thereby going beyond correlational associations.

On a technical note, it seems feasible that mildew-infested leaves would have been selected for extraction, but it is harder to explain how aphid DNA would have been extracted alongside plant DNA. Presumably, all leaves would have been cleaned of live aphids before they were placed in extraction tubes. What then is the origin of aphid DNA in these samples? Are these trace amounts from aphid saliva and faeces/honeydew that were left on the leaves? If this is the case, I would expect there to be substantially more mildew DNA than aphid DNA, yet the absolute read counts for aphids are actually higher. Presumably read counts should only be used as a relative metric within a pest organism, but this unexpected result nonetheless raises questions about what these read counts reflect. Again, having experimental data from different aphid densities would make these results more convincing.

Comments on revised version:

The authors have addressed many technical details in their revision, but they did not address my more fundamental concerns about validation of their results. I still believe that validation would be needed, but I also acknowledge that an additional experiment that reliably tests a causal relationship between read counts and pest abundance would go beyond the scope of a revision. Nonetheless, the authors currently only show variation in pest read counts among plant accessions, not in pest abundance. While the two measures are likely correlated, I hope that future studies will address more directly how pest abundance and read counts are causally linked, and whether pest read counts truly are a robust measure of pest abundance across a range of conditions and systems

---

## [Referee Report · Reviewer #2 (Public review)]

Summary:

Galanti et al investigate genetic variation in plant pest resistance using non-target reads from whole-genome sequencing of 207 field lines spontaneously colonized by aphids and mildew. They calculate significant differences in pest DNA load between populations and lines, with heritability and correlation with climate and glucosinolate content. By genome-wide association analyses they identify known defence genes and novel regions potentially associated with pest load variation. Additionally, they suggest that differential methylation at transposons and some genes are involved in responses to pathogen pressure. The authors present in this study the potential of leveraging non-target sequencing reads to estimate plant biotic interactions, in general for GWAS, and provide insights into the defence mechanisms of Thlaspi arvense.

Strengths:

The authors ask an interesting and important question. Overall, I found the manuscript very well-written, with a very concrete and clear question, a well-structured experimental design, and clear differences from previous work. Their important results could potentially have implications and utility for many systems in phenotype-genotype prediction. In particular, I think the use of unmapped reads for GWAS is intriguing.

Comments on revised version:

The revisions to the manuscript have significantly enhanced its clarity and scientific rigor. Methodological clarifications, especially regarding the normalization of read counts, now provide a stronger foundation for the presented results. Statistical enhancements, including more robust methods for controlling population structure and refined GWAS approaches, have solidified the reliability of the findings, effectively linking genetic variants and epigenetic modifications to pest loads. The discussion section has been improved to offer a more cautious interpretation of the correlations between transposable element (TE) methylation and pathogen load, emphasizing the associative nature of these findings. Additionally, increased transparency in data handling, particularly the treatment of ambiguous reads, has significantly reduced potential biases. These improvements have made the manuscript better suited to the readership, providing clearer insights into the genomic and epigenetic underpinnings of plant pest resistance.

---

## [Author Response]

The following is the authors’ response to the original reviews.

**Reviewer #1 (Public Review):**
Galanti et al. present an innovative new method to determine the susceptibility of large collections of plant accessions towards infestations by herbivores and pathogens. This work resulted from an unplanned infestation of plants in a greenhouse that was later harvested for sequencing. When these plants were extracted for DNA, associated pest DNA was extracted and sequenced as well. In a standard analysis, all sequencing reads would be mapped to the plant reference genome and unmapped reads, most likely originating from 'exogenous' pest DNA, would be discarded. Here, the authors argue that these unmapped reads contain valuable information and can be used to quantify plant infestation loads.For the present manuscript, the authors re-analysed a published dataset of 207 sequenced accessions of Thlaspi arvense. In this data, 0.5% of all reads had been classified as exogenous reads, while 99.5% mapped to the T. arvense reference genome. In a first step, however, the authors repeated read mapping against other reference genomes of potential pest species and found that a substantial fraction of 'ambiguous' reads mapped to at least one such species. Removing these reads improved the results of downstream GWAs, and is in itself an interesting tool that should be adopted more widely.The exogenous reads were primarily mapped to the genomes of the aphid Myzus persicae and the powdery mildew Erysiphe cruciferarum, from which the authors concluded that these were the likely pests present in their greenhouse. The authors then used these mapped pest read counts as an approximate measure of infestation load and performed GWA studies to identify plant gene regions across the T. arvense accessions that were associated with higher or lower pest read counts. In principle, this is an exciting approach that extracts useful information from 'junk' reads that are usually discarded. The results seem to support the authors' arguments, with relatively high heritabilities of pest read counts among T. arvense accessions, and GWA peaks close to known defence genes. Nonetheless, I do feel that more validation would be needed to support these conclusions, and given the radical novelty of this approach, additional experiments should be performed.A weakness of this study is that no actual aphid or mildew infestations of plants were recorded by the authors. They only mention that they anecdotally observed differences in infestations among accessions. As systematic quantification is no longer possible in retrospect, a smaller experiment could be performed in which a few accessions are infested with different quantities of aphids and/or mildew, followed by sequencing and pest read mapping. Such an approach would have the added benefit of allowing causally linking pest read count and pest load, thereby going beyond correlational associations.On a technical note, it seems feasible that mildew-infested leaves would have been selected for extraction, but it is harder to explain how aphid DNA would have been extracted alongside plant DNA. Presumably, all leaves would have been cleaned of live aphids before they were placed in extraction tubes. What then is the origin of aphid DNA in these samples? Are these trace amounts from aphid saliva and faeces/honeydew that were left on the leaves? If this is the case, I would expect there to be substantially more mildew DNA than aphid DNA, yet the absolute read counts for aphids are actually higher. Presumably read counts should only be used as a relative metric within a pest organism, but this unexpected result nonetheless raises questions about what these read counts reflect. Again, having experimental data from different aphid densities would make these results more convincing.

We agree with the reviewer that additional aphid counts at the time of (or prior to) sequencing would have been ideal, but unfortunately we do not have these data. However, compared to such counts one strength of our sequencing-based approach is that it (presumably) integrates over longer periods than a single observation (e.g. if aphid abundances fluctuated, or winged aphids visited leaves only temporarily), and that it can detect pathogens even when invisible to our eyes, e.g. before a mildew colony becomes visible. Moreover, the key point of our study is that we can detect variation in pest abundance even in the absence of count data, which are really time consuming to collect.

Conducting a new experiment, with controlled aphid infestations and continuous monitoring of their abundances, to test for correlation between pest abundance and the number of detected reads would require resequencing at least 30-50% of the collection for the results to be reliable. It would be a major experimental study in itself.

Regarding the origin of aphid reads and the differences in read-counts between e.g. aphids and mildew, we believe this should not be of concern. DNA contamination is very common in all kinds of samples, but these reads are simply discarded in other studies. For example, although we collected and handled samples using gloves, MG-RAST detected human reads (Hominidae, S2 Table), possibly from handling the plants during transplanting or phenotyping 1-2 weeks before sequencing. Therefore, although we did remove aphids from the leaves at collection, aphid saliva or temporary presence on leaves must have been enough to leave detectable DNA traces. Additionally, the fact that the *M. persicae* load strongly correlates with the *Buchnera aphidicola* load (R2=0.86, S6 Table), is reassuring. This obligate aphid symbiont is expected to be found in high amounts when sequencing aphids (see e.g. The International Aphid Genomics Consortium (2010))

The higher amount of aphid compared to mildew reads, can probably be explained by aphids having expanded more than mildew at the time of plant collection, but most importantly, as already mentioned by the reviewer, the read-counts were meant to compare plant accessions rather then pests to one another. We are interested in relative not absolute values. Comparisons between pest species are a challenge because they can be influenced by several factors such as the availability of sequences in the MG-RAST database and the DNA extraction kit used, which is plant-specific and might bias towards certain groups. All these potential biases are not a concern when comparing different plants as they are equally subject to these biases.

**Reviewer #2 (Public Review):**
Summary:Galanti et al investigate genetic variation in plant pest resistance using non-target reads from whole-genome sequencing of 207 field lines spontaneously colonized by aphids and mildew. They calculate significant differences in pest DNA load between populations and lines, with heritability and correlation with climate and glucosinolate content. By genome-wide association analyses they identify known defence genes and novel regions potentially associated with pest load variation. Additionally, they suggest that differential methylation at transposons and some genes are involved in responses to pathogen pressure. The authors present in this study the potential of leveraging non-target sequencing reads to estimate plant biotic interactions, in general for GWAS, and provide insights into the defence mechanisms of Thlaspi arvense.Strengths:The authors ask an interesting and important question. Overall, I found the manuscript very well-written, with a very concrete and clear question, a well-structured experimental design, and clear differences from previous work. Their important results could potentially have implications and utility for many systems in phenotype-genotype prediction. In particular, I think the use of unmapped reads for GWAS is intriguing.

Thank you for appreciating the originality and potential of our work.

Weaknesses:I found that several of the conclusions are incomplete, not well supposed by the data and/or some methods/results require additional details to be able to be judged. I believe these analyses and/or additional clarifications should be considered.

Thank you very much for the supportive and constructive comments. They helped us to improve the manuscript.

**Recommendations for the authors:**

**Reviewing Editor (Recommendations For The Authors):**
The authors address an interesting and significant question, with a well-written manuscript that outlines a clear experimental design and distinguishes itself from previous work. However, some conclusions seem incomplete, lacking sufficient support from the data, or requiring additional methodological details for proper evaluation. Addressing these limitations through additional analyses or clarifications is recommended.
**Reviewer #2 (Recommendations For The Authors):**
Major comments:- So far it is not clear to me how read numbers were normalised and quantified. For instance, Figure 1C only reports raw read numbers. In L149: "Prior to these analyses, to avoid biases caused by different sequencing depths, we corrected the read counts for the total numbers of deduplicated reads in each library and used the residuals as unbiased estimates of aphid, mildew and microbe loads". Was library size considered? Is the load the ratio between exogenous vs no exogenous reads? It is described in L461, but according to this, read counts were normalised and duplicated reads were removed. Now, why read counts were used? As opposite to total coverage / or count of bases per base? I cannot follow how variation in sequencing quality was considered. I can imagine that samples with higher sequencing depth will tend to have higher exogenous reads (just higher resolution and power to detect something in a lower proportion).

Correcting for sequencing depth/library size is indeed very important. As the reviewer noted, we had explained how we did this in the methods section (L464), and we now also point to it in the results (L151):

“Finally, we log transformed all read counts to approximate normality, and corrected for the total number of deduplicated reads by extracting residuals from the following linear model, log(read_count + 1) ∼ log(deduplicated_reads), which allowed us to quantify non-Thlaspi loads, correcting for the sequencing depth of each sample.”

We showed the uncorrected read-counts only in Fig 1 to illustrate the orders of magnitude but used the corrected read-counts (also referred to as “loads”) for all subsequent analyses.

In our view, theoretically, the best metric to correct the number of reads of a specific contaminant organism, is the total number of DNA fragments captured. Importantly, this is not well reflected by the total number of raw reads because of PCR and optical duplicates occurring during library prep and sequencing. For this reason we estimated the total number of reads captured multiplying total raw reads (after trimming) by the deduplication rate obtained from FastQC (methods L409-411). This metric reflects the amount of DNA fragments sampled better than the raw reads. Also it better reflects MG-RAST metrics as this software also deduplicates reads (Author response image 1 below). We also removed duplicates in our strict mappings to the *M. persicae* and *B. aphidicola* genomes.

Coverage is not a good option for correction, because it is defined for a specific reference genome and many of the read-counts output by MG-RAST do not have a corresponding full assembly. Moreover, coverage and base counts are influenced by read size, which depends on library prep and is not included in the read-counts produced by MG-RAST.

**Author response image 1. sa3fig1:** Linear correlations between the number of MG-RAST reads post-QC and either total (left) or deduplicated (right) reads from fastq files of four full samples (not only unmapped reads).

- The general assumption is that plants with different origins will have genetic variants or epigenetic variations associated with pathogen resistance, which can be tracked in a GWAS. However, plants from different regions will also have all variants associated with their origin (isolation by state as presented in the manuscript). In line 169: "Having established that our method most likely captured variation in plant resistance, we were interested in the ecological drivers of this variation". It is not clear to me how variation in plant resistance is differentiated from geographical variation (population structure). in L203: "We corrected for population structure using an IBS matrix and only tested variants with Minor Allele Frequency (MAF) > 0.04 (see Methods).". However, if resistant variants are correlated with population structure as shown in Table 1, how are they differentiated? In my opinion, the analyses are strongly limited by the correlation between phenotype and population structure.

The association of any given trait with population structure is surely a very important aspect in GWAS studies and when looking at correlations of traits with environmental variables. If a trait is strongly associated with population structure, then disentangling variants associated with population structure vs. the ones associated with the trait can indeed be challenging, a good example being flowering time in *A. thaliana* (e.g. Brachi et al. 2013).

In our case, although the pest and microbiome loads are associated with population structure to some extent, this association is not very strong. This can be observed for example in Fig. 1C, where there is no clear separation of samples from different regions. This means that we can correct for population structure (in both GWAS and correlations with climatic variables) without removing the signals of association. It is possible that other associations were missed if specific variants were indeed strongly associated with structure, but these would be unreliable within our dataset, so it is prudent to exclude them.

- Similarly, in L212: "we still found significant GWA peaks for Erysiphales but not for other types of exogenous reads (excluding isolated, unreliable variants) (Figure 3A and S3 Figure)." In a GWA analysis, multiple variants will constitute an association pick (as shown for instance in main Figure 3A) only when the pick is accentuated by lockage disequilibrium around the region under selection (or around the variant explaining phenotypic variation in this case). However, in this case, I suspect there is a strong component of population structure (which still needs to be corroborated as suggested in the previous comment). But if variants are filtered by population structure, the only variants considered are those polymorphic within populations. In this case, I do not think clear picks are expected since most of the signal, correlated with population has been removed. Under this scenario, I wonder how informative the analyses are.

As mentioned above, the traits we analyse (aphid and mildew loads) are only partially associated with population structure. This is evident from Fig. 1C (see answer above) but also from the SNP-based heritability (Table 1, last column) which measures indeed the proportion of variance explained by genetic population structure. Although some variance is explained (i.e. the reviewer is correct that there is some association) there is still plenty of leftover variance to be used for GWAS and correlations with environmental variables. The fact that we still find GWAS peaks confirms this, as otherwise they would be lost by the population structure correction included in our mixed model.

- How were heritability values calculated? Were related individuals filtered out? I suggest adding more detail in both the inference of heritability and the kinship matrix (IBS matrix). Currently missing in methods (for heritability I only found the mention of an R package in the caption of Table 1).

We somehow missed this in the methods and thank the reviewer for noticing. We now added this paragraph to the chapter “Exogenous reads heritability and species identification”:

“To test for variation between populations we used a general linear model with population as a predictor. To measure SNP-based heritability, i.e. the proportion of variance explained by kinship, we used the marker_h2() function from the R package heritability (Kruijer and Kooke 2019), which uses a genetic distance matrix as predictor to compute REML-estimates of the genetic and residual variance. We used the same IBS matrix as for GWAS and for the correlations with climatic variables.”

We also added the reference to the R package *heritability* to the Table 1 caption.

- Figure 2C. in line 188: "Although the baseline levels of benzyl glucosinolates were very low and probably sometimes below the detection level, plant lines where benzyl glucosinolate was detected had significantly lower aphid loads (over 70% less reads) in the glasshouse (Figure 3C)". It is not clear to me how to see these values in Figure 2C. From the boxplot, the difference in aphid loads between detected and not detected benzyl seems significantly lower. From the boxplot distribution is not clear how this difference is statistically significant. It rather seems like a sampling bias (a lot of non-detected vs low detected values). Is the difference still significant when random subsampling of groups is considered?

Here the “70% less reads” refers to the uncorrected read-counts directly (difference in means between samples where benzyl-GS were detected vs. not). We agree with the reviewer that this is confusing when referred to figure 2C which depicts the corrected *M. persicae* load (residuals). We therefore removed that information.

Regarding the significance of the difference, we re-calculated the p value with the Welch's t-test, which accounts for unequal variances, and with a bootstrap t-test. Both tests still found a significant difference. We now report the p value of the Welch’s t-test.

- I think additional information regarding the read statistics needs to be improved. At the moment some sections are difficult to follow. I found this information mainly in Supplementary Table 1. I could not follow the difference in the manuscript and supplementary materials between read (read count), fragment, ambiguous fragments, target fragments, etc. I didn't find information regarding mean coverage per sample and relative plant vs parasite coverage. This lack of clarity led me to some confusion. For instance, in L207: "We suspected that this might be because some non-Thlaspi reads were very similar to these highly conserved regions and, by mapping there, generated false variants only in samples containing many non-Thlaspi reads". I find it difficult to follow how non-Thlaspi reads will interfere with genotyping. I think the fact that the large pick is lost after filtering reads is already quite insightful. However, in principle I would expect the relative coverage between non-Thlaspi:Thlaspi reads to be rather low in all cases. I would say below 1%. Thus, genotyping should be relatively accurate for the plant variants for the most part. In particular, considering genotyping was done with GATK, where low-frequency variants (relative coverage) should normally be called reference allele for the most part.

We agree with the reviewer that some clarification over these points is necessary! We modified Supplementary Table 1 to include coverage information for all samples before and after removal of ambiguous reads and explained thoroughly how each value in the table was obtained. Regarding reads and fragments, we define each fragment as having two reads (R1 and R2). The classification into Target, Ambiguous and Unmapped reads was based on fragments, so we used that term in the table, but referring to reads has the same meaning in this context as for example an unmapped read is a read whose fragment was classified as unmapped.

We did not include the pest coverage specifically, because this cannot be calculated for any of the read counts obtained with MG-RAST as this tool is mapping to online databases where genome size is not necessarily known. What is more meaningful instead are the read counts, which are in Supplementary tables 2 and 6. Importantly as mentioned in other answers, if different taxa are differently represented in the databases this does not affect the comparison of read counts across different samples, but only the comparison of different taxa which was not used for any further analyses.

Regarding the ambiguous reads causing unreliable variants, these occur only in very few regions of the Thlaspi genome that are highly conserved in evolution or of very low complexity. In these regions reads generated from both plant or for instance aphid DNA, can map, but the ones from aphid might contain variants when mapping to the Thlaspi reference genome (L207 and L300). The reviewer is right that there is only a very small difference in average coverage when removing those ambiguous reads (~1X, S1 Table), but that is not true for those few regions where coverage changes massively when removing ambiguous reads as shown on the right side Y axes of S2 Figure. Therefore these unreliable variants are not low-frequency and therefore not removed by GATK.

- L215. I am not very convinced with the enrichment analyses, justified with a reference (52). For instance, how many of the predicted picks are not close to resistance genes? How was the randomisation done? At the moment, the manuscript reads rather anecdotally by describing only those picks that effectively are "close" to resistance genes. For instance, if random windows (let's say 20kb windows) are sampled along the genome, how often there are resistant genes in those random windows, and how is the random sampling compared with observed picks (windows).

Enrichment is by definition an increase in the proportion of true positives (observed frequency: proportion of significant SNPs located close to a priori candidate genes) compared to the background frequency (number of all SNPs located close to a priori candidate genes). So the background likelihood of SNPs to fall into a priori candidate SNPs (i.e. the occurrence of a priori candidate genes in randomly sampled windows, as suggested by the reviewer) is already taken into account as the background frequency. We now explained more extensively how enrichment is calculated in the relevant methods section (L545-549), but it is an extensively used method, established in a large body of literature, so it can be found in many papers (e.g. Atwell et al. 2010, Brachi et al. 2010, Kawakatsu et al. 2016, Kerdaffrec et al. 2017, Sasaki et al. 2015-2019-2022, Galanti et al. 2022, Contreras-Garrido et al. 2024).

Although we had already calculated an upper bound for the FDR based on the a priori candidates, as in previous literature, we now further calculated the significance of the enrichment for the Bonferroni-corrected -log(p) threshold for Erysiphales. Calculating significance requires adopting a genome rotation scheme that preserves the LD structure of the data, as described in the previously mentioned literature (eg. Kawakatsu et al. 2016, Sasaki et al. 2022). Briefly, we calculated a null distribution of enrichments by randomly rotating the p values and a priori candidate status of the genetic variants within each chromosome, for 10 million permutations. We then assessed significance by comparing the observed enrichment to the null distribution. We found that the enrichment at the Bonferroni corrected -log(p) threshold is indeed significant for Erysiphales (*p* = 0.016). We added this to the relevant methods section and the code to the github page.

In addition, many other genes very close (few kb max) to significant SNPs were not annotated with the “defense response” GO term but still had functions relatable to it. Some examples are *CAR8*, involved in ABA signalling, *PBL7* in stomata closure and *SRF3* in cell wall building and stress response (Fig 3D). This means that our enrichment is actually most likely underestimated compared to if we had a more complete functional annotation.

- L247. Additional information is needed regarding sampling. It is not clear to me why methylation analyses are restricted to 20 samples, contrary to whole genome analyses.

The sampling is best described in the original paper (on natural DNA methylation variation; Galanti et al. 2022), although the most important parts are repeated in the first chapter of the methods.

Regarding methylation analysis, they are not restricted to 20 samples. Only the DMR calling was restricted to the 20 vs. 20 samples with the most divergent values (of pest loads) to identify regions of variation. This analysis was used to subset the genome to potential regions associated with pest presence rather than thoroughly testing actual methylation variants associated with pest presence. The latter was done in the second step, EWAS, which was based on the whole dataset with the exclusions of samples with high non-conversion rate. This left 188 samples for EWAS. We added this number in the new manuscript (L251 and L571).

To clarify, we made a few additions to the results (L250) and methods (last two subchapters) sections, where we explain the above.

- No clear association with TEs: in L364: "Erysiphales load was associated with hypomethylated Copia TEs upstream of MAPKKK20, a gene involved in ABA-mediated signaling and stomatal closure. Since stomatal closure is a known defense mechanism to block pathogen access (21), it is tempting to conclude that hypomethylation of the MAPKKK20 promoter might induce its overexpression and consequent stomatal closure, thereby preventing mildew access to the leaf blade. Overall, we found associations between pathogen load and TE methylation that could act both in cis (eg. Copia TE methylation in MAPKKK20 promoter) and in trans, possibly through transposon reactivation (eg. LINE, Helitron, and Ty3/Gypsi TEs isolated from genes)." I find the whole discussion related to transposable elements, first, rather anecdotical, and second very speculative. To claim: "Overall, we found associations between pathogen load and TE methylation", I believe a more detailed analysis is needed. For instance, how often there is an association? In general, there are some rather anecdotical examples, several of which are presented as association with pathogen load on the basis of being "in proximity" to a particular region/pick. The same regions contain multiple other genes and annotations, but the authors limit the discussion to the particular gene or TE concordant with the hypothesis. This is for both the discussion and results sections.

Here we are referring to associations in a purely statistical sense. The fact that “Overall, we found associations between pathogen load and TE methylation” is simply a conclusion drawn from Fig. 4b, without implying any causality. Some methylation variants are statistically associated with the traits (aphid or mildew loads), and whether they are true positives or causal is of course more difficult to assess.

Regarding the methylation variants associated with mildew load in proximity of *MAPKKK20*, those are the only two significant ones, located close to each other and close to many other variants that, although not significant, have low *P*-values (Author response image 2 below), so it is the most obvious association warranting further exploration. The reviewer is correct that there are other genes flanking the large DMR that covers the TEs (Fig. 4D), but the DMR is downstream of these genes, so less likely to affect their transcription.

**Author response image 2. sa3fig2:** 

Regarding all other associations found with *M. persicae* load, we stated that these are not really reliable due to a skewed *P*-value distribution (L269, S5B Fig), but we think that for future reference it is still worth reporting the closeby genes and TEs.

We slightly changed the wording of the passage the reviewer is citing above to make it clearer that we are only offering potential explanations for the associations we observe with TE methylation, but by no means we state that TE reactivation is surely what is happening.

- One conclusion in the manuscript is that DMRs have been mostly the result of hypomethylation. This is shown for instance in supplementary Figure 4. However, no general statistic is shown of methylation distribution (not only restricted to DMRs). Was the ratio methylation over de-methylation proportional along the genome? Thus the finding in DMRs is out of the genome-wide distribution? Or on the contrary, the DMRs are just a random sampling of the global distribution. The same for different annotated regions. For instance, I would expect that in general coding regions would be less methylated (not restricted to DMRs).

Complete and exhaustive analyses of the methylomes were already published in the original manuscript (Galanti et al 2022). However, the variation among these methylomes is complex and influenced by multiple factors including genetic background and environment of origin, and talking about these things would have been beyond the scope of our paper. In this paper, we just took advantage of the existing methylome information to identify the few genomic regions that are consistently differentially methylated between samples with extreme values of pest loads. As for the GWAS, the phenotypes are only partially associated with population structure, so the 20 samples with the lowest and the 20 with the highest pathogen loads are not e.g. all Swedish vs. all German but they are a mixture, which allowed us to correct for population structure running EWAS with a mixed model that includes a genetic distance matrix.

In this study we called DMRs between two defined groups: samples with the lowest amounts of pathogen DNA (not-infected; the “control” group) vs. samples with the highest amounts of pathogens (infected or the “treatment” group), so we could define a directionality (“hyper vs. “hypo” methylation). However, this is not the case for population DMRs called between many different combinations of populations. This is why the hyper- and hypomethylated regions found here cannot be compared to the genome-wide averages, which are influenced by other factors than the pathogens. Even with relaxed thresholds we indeed found very few DMRs associated to pathogen presence here.

Specifically about coding regions, the reviewer is correct that they are less methylated, especially because *T. arvense* has largely lost gene body methylation (Nunn et al. 2021, Galanti et al. 2022), but this is unrelated and was discussed in the original publication (Galanti et al. 2022).

Minor comments:- Figure 1B: it would be good to add also percentage values.

As the figure is already tightly packed, we rather keep it simple. As the chart gives a good impression of frequencies of different kingdoms, and the frequences of several relevant groups. Also, as explained in a previous answer, comparing different taxonomic groups could be imprecise (as opposed to comparing the same group between different samples), so exact percentages seem unnecessary. If needed, the exact percentages can still be calculated from S2 Table.

- L159: It is not clear to me what "enemy variation" is referring to here.

We are referring to variation in enemy densities (attack rates) in the field, that could potentially be carried over to the greenhouse to cause the patterns of infection we observed. We changed it to “variation in enemy densities” to make it more clear.

- L259: "In accordance with previous studies (8,9), most DMRs were hypomethylated in the affected samples, indicating that genes needed for defense might be activated through demethylation". Not clear to me what "affected samples" is referring to. Samples with lower load?

Affected samples have a higher load of pathogen reads. We changed it to “infested” to make it more clear.

- L336. Figure should be Fig 3E.

We fixed it, thanks for noticing.

ADDITIONAL CHANGES

We updated reference 43 to point to the published paper rather than the preprint.

We corrected the phenotype names in S3 Fig, to make them consistent with the rest of the manuscript and increased font size on the axes to make it more readable.